# Machine Learning in Stream/River Water Temperature Modeling: a review and metrics for evaluation

Claudia R. Corona[1], Terri S. Hogue[1,2]

[1]Department of Civil and Environmental Engineering, Colorado School of Mines, Golden, 80401, United States
[2]Hydrologic Science and Engineering Program, Colorado School of Mines, Golden, 80401, United States

*Correspondence to*: Claudia R. Corona (claudia.corona@mines.edu)

**Abstract.** As climate change continues to affect stream/river (henceforth stream) systems worldwide, stream water temperature (SWT) is an increasingly important indicator of distribution patterns and mortality rates among fish, amphibians, and macroinvertebrates. Technological advances tracing back to the mid-20th century have improved our ability
to measure SWT at varying spatial and temporal resolutions for the fundamental goal of better understanding stream function and ensuring ecosystem health. Despite significant advances, there continue to be numerous stream reaches, stream segments and entire catchments that are difficult to access for a myriad of reasons, including but not limited to physical limitations. Moreover, there are noted access issues, financial constraints, and temporal and spatial inconsistencies or failures with in-situ instrumentation. Over the last few decades and in response to these limitations, statistical methods and physically based
computer models have been steadily employed to examine SWT dynamics and controls. Most recently, the use of artificial intelligence, specifically machine learning (ML) algorithms, has garnered significant attention and utility in hydrologic sciences, specifically as a novel tool to learn undiscovered patterns from complex data and try to fill data streams and knowledge gaps. Our review found that in the recent five years (2020-2024), more studies using ML for SWT were published, than had been in the previous 20 years, (2000-2019), totaling 57. The aim of this work is three-fold: first, to
provide a concise review of the use of ML algorithms in SWT modeling and prediction, second, to review ML performance evaluation metrics as it pertains to SWT modeling and prediction and find the commonly used metrics and suggest guidelines for easier comparison of ML performance across SWT studies and third, to examine how ML use in SWT modeling has enhanced our understanding of spatial and temporal patterns of SWT and examine where progress is still needed.

# 1 Introduction

Water temperature in a stream/river plays a vital role in nature and society, regulating dissolved oxygen concentrations (Poole & Berman, 2001), biochemical oxygen demand rates, and chemical toxicities (Cairns et al., 1975; Patra et al., 2015). Additionally, SWT is an important indicator of cumulative anthropogenic impacts on lotic environments (Risley et al., 2010). Observations of SWT changes over time can reveal the effects of stream flow regulation, riparian alteration (Johnson & Jones, 2000), and large-scale climate change (Barbarossa et al., 2021) on local ecosystems. From an ecological standpoint, SWT strongly influences (Ward, 1998) the health, survival, and distribution of freshwater fish (Ulaski et al., 2023; Wild et al., 2023), amphibians (Rogers et al., 2020) and macroinvertebrates (Wallace & Webster, 1996). As climate change progresses, SWT will be an increasingly critical proxy for ecosystem health and function, both locally and nationally.

## 1.1 SWT modeling in the 21st century

Technological advances since the turn of the 20th century have improved our ability to measure SWT in an affordable and dependable manner at varying spatial and temporal resolutions (Benyahya et al., 2007; Dugdale et al., 2017). Despite significant advances in the last 100 years, there remain many stream reaches, stream segments and entire catchments that are difficult to access for a myriad of reasons (Ouellet et al., 2020), including, but not limited to physical limitations: i.e., streams may be in private property, remote or dangerous-to-access areas, financial constraints: access may be limited by monetary resources or lack-there-of, or temporal limitations such as uncertainties/inconsistencies in the continuity of measurements or unforeseen equipment loss/failure (Webb et al., 2015; Isaak et al., 2017). In response to these limitations, statistical methods and physically based computer models have been steadily employed over the last few decades to support the advancement of scientific understanding of stream form and function and subsequent implications for water management (Cluis, 1972; Caissie et al., 1998; Mohseni-Astani et al., 2010; DeWeber & Wagner, 2014; Isaak et al., 2017).

Aided by the continued development of computers and the internet, physical and statistical computer models have gained prominence outside of academia and are more commonly being used by stakeholders and local groups to address a myriad of hydrology challenges (Maheu et al., 2016; Liu et al., 2018; Tao et al., 2020; Rogers et al., 2020). At the same time, the problem-solving success of machine learning (ML), which falls under the umbrella of artificial intelligence, has become increasingly popular in hydrologic sciences in the last few years (DeWeber & Wagner, 2014; Xu & Liang, 2021). Artificial intelligence (AI) describes technologies that can incorporate and assess inputs from an environment, learn optimal patterns, and implement actions to meet stated objectives or performance metrics (Xu & Liang, 2021; Varadharajan et al., 2022). As a subset of AI, the goal of ML algorithms and models is to learn patterns from complex data (Friedberg, 1958). A global call to better predict and prepare for near- and far-future hydrologic conditions has led researchers in the last few decades to use ML algorithms to model hydrologic processes at various temporal and spatial scales (Poff et al., 1996; Solomatine et al., 2008; Cole et al., 2014; Khosravi et al., 2023). For example, a type of ML called artificial neural networks (ANNs), have been used since the 1990s in many subfields of hydrology, such as streamflow predictions (Karunanithi et al., 1994; Poff et

al., 1996), rainfall-runoff modeling (Hsu et al., 1995; Shamseldin, 1997), subsurface flow and transport (Morshed & Kaluarachchi, 1998), and flood forecasting (Thirumalaiah & Deo, 1998). For SWT modeling however, the use of ML algorithms such as ANNs have only recently garnered interest (Zhu & Piotrowski, 2020).

## 1.2. Study Objective

The current work includes an extensive literature review of studies that used ML algorithms/models for river/SWT modeling, hindcasting and forecasting. The intent of this review is two-fold: 1) to introduce ML for hydrologists who have modeling experience and are interested in pursuing ML-use for their SWT studies, and 2) to provide a broad overview of machine learning applications in SWT. For ML experts, we think that this review could also prove useful as reference for how ML has been applied in the field of SWT modeling and where improvement is needed. Overall, this article aims to serve as a bridge between hydrologists and machine learning experts. Our review includes papers cited by Zhu and Piotrowski (2020), who previously conducted a study of ANNs used in SWT modeling, however, we provide a comprehensive examination of peer-reviewed journals that use any type of artificial intelligence/ML algorithm to model or evaluate river/SWT. This review's first objective is to provide a concise review of ML algorithm use in SWT modeling. Secondly, our goal is to examine the ML performance evaluation metrics used in SWT modeling and find the most-used metrics and suggest guidelines for clearer comparison of ML performance. The third objective is to discuss the community's use of ML to address physical system understanding in SWT modeling. Overall, this review aims to serve as a critical assessment of the state of SWT understanding given the increasing popularity of ML use in SWT modeling.

## 2 Overview: Stream Water Temperature Model Types

### 2.1 SWT statistical (also stochastic or empirical) models

In the 1960s, considerable interest grew in the prediction of SWT, particularly in the western United States (U.S.) due to increased awareness of environmental quality issues (Ward, 1963; Edinger et al., 1968; Brown, 1969). The creation of large dams, daily release of heated industrial effluents, growing agricultural waste discharge and forest clear-cutting could influence downstream SWT. However, the extent of such influence remained poorly understood and difficult to test at large spatial and temporal scales (Brown, 1969). From the 1960s to the 1970s, understanding of the relationship between SWT and ambient air temperature (AT) was solidified, and scientists began to increasingly use statistical methods to examine the air-water relationships in stream environments (Morse, 1970; Cluis, 1972). Statistical (also stochastic or empirical) models are governed by empirical relations between SWT and their predictors, which requires fewer input data. An example of such progress took place in Canada, where researchers created an autoregressive model to calculate mean daily SWT fluctuations using six months of data from the summer and winter months of 1969 (Cluis, 1972). Cluis (1972) further said that their model was transferrable to other streams of comparable size. The use of statistical methods in SWT modeling became increasingly common in the latter half of the 20$^{th}$ century due in large part to minimal data requirements (Benyahya et al.,

2007). For example, scientists in Europe used limited data and statistics to examine the influence of atmospheric and topographic factors on the temperature of a small upland stream (Smith & Lavis, 1975). In Australia, scientists interested in finding limits for reaches of streams downstream from thermal discharges, found a simple method that could predict SWT based solely on-site altitude and AT or upstream SWT (Walker & Lawson, 1977). In Canada, SWT was predicted using a stochastic approach, which included the use of Fourier series, multiple regression analysis, Markov processes, and a Box-Jenkins time-series model (Caissie et al., 1998). In the 21st century, statistical methods continue to be a prominent tool used for SWT modeling and prediction (Ahmadi-Nedushan et al., 2007; Chang & Psaris, 2013; Segura et al., 2015; Detenbeck et al., 2016; Siegel & Volk, 2019; Ulaski et al., 2023; Fuller et al., 2023). We refer the reader to Benyahya et al. (2007) for a comprehensive review of SWT statistical models and approaches.

## 2.2 SWT physically based (also process-based, deterministic, mechanistic) models

While statistical methods can be straightforward to use and requires minimal in situ data for first analysis (Benyahya et al., 2007), limitations and uncertainty with regards to SWT predictions is possible, specifically when trying to understand the controls of energy transfer mechanisms responsible for trends (Dugdale et al., 2017). To address these shortcomings and with the introduction of personal computers in the late 1960s (Dawdy & Thompson, 1967), researchers developed computer models and software programs that tried to address the more fundamental hydrology questions founded in physics and natural processes (Theurer et al., 1985; Bartholow, 1989). One example of such progress was a SWT prediction one-dimensional computer model that used a simplified energy conservation equation to predict SWT for the upper reaches of the Columbia River in the Pacific Northwest of the U.S., during July 1966 (Morse, 1970). These models are described as being physically based or process-based (alternatively called 'deterministic' or 'mechanistic' models).

Due to the continued lack of sufficient in situ observations and resources with which to undertake field studies in SWT science (Dugdale et al., 2017), physically based models became increasingly used. From the end of the 20th century through the present, they are considered one of the best available options in generating predictions of SWT, particularly at a localized scale (Dugdale et al., 2017). Physically based models became useful enough that government agencies introduced their own models to encourage uniformity. In the 1980s, the U.S. Geological Survey (USGS) introduced a physically based model that simulated SWT called SNTemp (Theurer et al., 1985; Bartholow, 1989). A few years later, the U.S. Environmental Protection Agency (EPA) introduced SHADE-HSPF for similar purposes (Chen, Carsel, et al., 1998; Chen, McCutcheon, et al., 1998). Where available, academic scientists coupled field measurements with physically based numerical models. For example, scientists in Minnesota created a numerical model, called MNSTREM, based on a finite difference solution of the nonlinear equation to predict SWT at one-hour increments for the Clearwater River (Sinokrot & Stefan, 1993). Similarly, academic scientists in Canada introduced CEQUEAU, a water-balance type model which incorporated vegetation and soil characteristics to solve for SWT (St-Hilaire et al., 2000). Physically based models became commercially available in the 2000s, one example being the MIKE suite of models, which were created to solve the heat and advection-dispersion equation to simulate both surface and subsurface water dynamics, created by the DHI consulting group (Jaber & Shukla, 2012; Loinaz

et al., 2013). In addition to the models mentioned, over a dozen more physically based models were created and used between 1990 and 2017 (Dugdale et al., 2017). For a more detailed review of physically based SWT models, we refer the reader to Dugdale et al. (2017).

## 2.3 Artificial Intelligence Models in SWT modeling

Initial discussion of artificial intelligence can be traced back to 1943, when McCulloch and Pitts presented a computer model that functioned like neural networks of the brain (McCulloch & Pitts, 1943). In 1958, R.M. Friedberg published "A Learning Machine: Part 1", in IBM's Journal of Research and Development, one of the first to describe the concept of "machine learning". Friedberg hypothesized that machines could be taught how to learn such that they developed the capability to improve their own performance to the point of completing tasks or meeting objectives (Friedberg, 1958). Sixty years later, ML has grown as a field of study in academia and as an area of great interest in society, the latter due in large part to the popularity of large language models (a type of machine learning that we will not discuss here), such as ChatGPT (OpenAI, Inc., 2024), Copilot (Microsoft, Inc., 2024) and Gemini (Google LLC, 2024).

In the last decade, computing advances in AI have started to offer several advantages for using machine learning (ML) in hydrology that are comparable to physically based models (Cole et al., 2014; Zhu et al., 2019; Rehana and Rajesh, 2023). In contrast to traditional physically based models, the code underlying ML models are generally open-source and publicly available allowing for near real-time accessible advances and user feedback, whereas the source code for some physically based models may be inaccessible to the public due to being privately managed (MIKE suite of models) or the model software may be publicly available but could take years to publish updates (USGS MODFLOW, Simunek's HYDRUS). One advantage that has made ML increasingly appealing includes its ability to learn directly from the data (i.e., data driven), which can be useful when the underlying physics are not fully understood or are considered too complex to model accurately.

Additionally, ML models are more efficient in making predictions compared to the time-intensive solvers of physically based models. ML models can also handle the challenge of scalability, that is managing large datasets and seamlessly deploying across various computer platforms and applications (Rehana and Rajesh, 2023). Air2stream, a hybrid statistical-physically based SWT model (Toffolon and Piccolroaz, 2015; Piccolroaz et al., 2016), initially outperformed earlier ML models such as Gaussian Process Regression (Zhu et al., 2019). However, in the last few years, Air2stream has had its performance matched and even exceeded by recent neural networks models (Feigl et al., 2021; Rehana and Rajesh, 2023). Finally, with computer processing power improving and the emergent field of quantum computing, there is a strong belief that using ML and by extension AI, in science applications will drive innovation to the point where natural patterns and insights not currently apparent in physical modeling will be uncovered (Varadharajan et al., 2022). Thus, while physically based models are considered invaluable for their interpretability and grounding in established physics, ML models have the potential for growth in various fields of hydrology, where they can be used to first complement and eventually lead as powerful tools for prediction, optimization, and understanding in increasingly complex and data-rich environments.

For this review, we differentiate between traditional ML and newer ML, where the former includes approaches that have been used for decades in hydrologic modeling, i.e., cluster analysis, support vector machine, and shallow neural networks. We define newer ML as those introduced in hydrologic modeling in recent years, such as the deep learning long short-term memory NNs, extreme learning machine, and ML hybridizations. The following sections provide an overview of ML types and learning techniques. Finally, we assume that readers have a very basic understanding of the differences between machine

learning types such as: supervised, semi-supervised, and unsupervised learning, and refer the reader to Xu and Liang (2021) for a nice overview.

### 2.3.1  Traditional ML algorithms

#### 2.3.1.1 K-nearest neighbors

*K-nearest neighbours* (K-nn) is a versatile supervised ML algorithm (Fix & Hodges, 1952; Cover & Hart, 1967) used to

solve nonparametric classification and regression problems. The K-nn algorithm uses proximity between data points to make classifications or evaluations about the grouping of any given data point. K-nn gained popularity in the 2000s due to its simplicity in implementation and understanding, making it readily accessible to hydrologic researchers and practitioners. For example, St.-Hilaire et al. (2011) used various K-nn model configurations to model SWT for the Moisie River in northern Quebec, Canada, finding that the best K-nn model required prior-day SWT data and day-of-year (DOY), an

indicator of seasonality. Advantages of K-nn include its non-assumptions of the underlying distribution of the data, allowing it to handle nonlinear complexities without requiring a solid model structure as is the case for some physical models (St. Hilaire et al., 2011). Disadvantages of K-nn are that it is computationally intensive, may require extensive cross-validation, performance can be affected by irrelevant/redundant features, and due to its high memory and computational needs, is impractical for large-scale applications, i.e., scalability issues, (Acito, 2023). For example, Heddam et al. (2022) compared

K-nn with other ML algorithms, finding that K-nn was outperformed by other MLs such as least squares support vector machine and neural networks. The use of K-nn may still be reasonable for simple, local cases but we advise other MLs for more complex or larger-use cases.

#### 2.3.1.2 Cluster analysis and variants

*Cluster analysis* is a category of unsupervised ML methods used to create groups from an unlabeled dataset. Clustering

methods use distance functions such as Euclidean distance, Manhattan distance, Minkowski distance, Cosine similarity and others, to group data into clusters (Irani et al., 2016). The analysis separates data into groups of maximum similarity, while also trying to minimize the similarity from group to group (Xu & Liang, 2021). In SWT modeling, studies have used cluster analysis to try a reduction of a dataset prior to assessment (Voza & Vuković, 2018) and/or to find spatiotemporal patterns in a dataset (Krishnaraj & Deka, 2020). Another popular clustering technique is discriminant analysis, which tries to find

parameters that are most significant for temporal differentiation between rendered periods (Voza & Vuković, 2018). *K-means*, a type of unsupervised ML, is a clustering algorithm that finds 'k' number of centroids in the dataset and distributes each respective data value to the nearest cluster while keeping the smallest number of centroids possible (Krishnaraj & Deka,

2020). Krishnaraj and Deka (2020) used K-means to organize spatial grouping for water quality monitoring stations for dry and wet regions along the Ganga river basin in India to identify whether pollution patterns could be discerned.

While cluster analysis and discriminant analysis are generally used to reduce datasets, another technique, the *Principal Component Analysis/factor test (PCA)*, is applied to assess dominant factors in datasets. Mathematically, Principal Component Analysis (PCA) is a statistical unsupervised ML technique that uses an orthogonal transformation (a linear transformation that preserves lengths of vectors and angles) to convert a set of variables from correlated to uncorrelated (Krishnaraj & Deka, 2020). Using PCA, Krishnaraj and Deka (2020) found that certain water quality parameters (dissolved

oxygen, sulfate, electrical conductivity) were more dominant in the dry season compared to the wet season (total dissolved solids, sodium, potassium, sodium, chlorine, chemical oxygen demand), data which could be used to cater the monitoring program to the important parameters. SWT was not a dominant parameter, likely in part because the SWT of large downstream rivers like the Gangas are generally less variable due to their larger volume and stronger thermal buffer.

### 2.3.1.3 Support vector machine and regression

*Support vector machine (SVM)* is a supervised learning technique used for classification, regression and outlier detection. The aim of SVM is to find a hyperplane (or the decision surface) in an N-dimensional space (N = number of features) that best separates labeled categories, or support vectors (Cortes & Vapnik, 1995). One of the advantages of SVM is that it seeks to minimize the upper bound of the generalization error, instead of the training error (Cortes & Vapnik, 1995). A big disadvantage is that it does not perform well with large data sets due to the likelihood of greater noise, which would cause

support vectors to overlap, making classification difficult. For a more detailed explanation of SVM, we refer the reader to Cortes & Vapnik (1995) and Xu and Liang (2021). In the last few decades, SVM has been coupled with other ML models to find the best performing models for short-term water quality predictions (Lu & Ma, 2020) and daily SWT modeling (Heddam, Ptak, et al., 2022). For example, Heddam (2022), used *Least Squares SVM (LSSVM)*, a variant of SVM which takes a linear approach (instead of quadratic like SVM) to reach a solution (Suykens & Vandewalle, 1999).

A version of SVM used for regression tasks is the *support vector regression (SVR)*. SVR attempts to minimize the objective function (composed of loss greater than a specified threshold) and a regularization term (Rehana, 2019; Hani et al., 2023). For further detail on SVR, we refer the reader to Rehana (2019) and Hani et al., (2023). Using historical data, SVR has been compared with other ML models that evaluate SWT variability due to climate change (Rehana, 2019), finding temperature increases less pronounced in the SVR model. Jiang et al. (2022) compared SVR to other ML models to forecast

SWT in cascade reservoir-influenced rivers. For the cascade reservoir operation-influenced study, SVR was outperformed by random forest (RF) and gradient boosting (Jiang et al., 2022). Focusing on 78 catchments in the Mid-Atlantic and Pacific Northwest hydrologic regions of the U.S., researchers used SVR, and a ML algorithm called XGBoost to predict monthly SWT (Weierbach et al., 2022), finding that SVR significantly outperformed traditional statistical approaches such as multi-linear regression (MLR), but did not outperform XGBoost. In addition, the SVR models had the highest accuracy for SWT

across different catchments (Weierbach et al., 2022). In Quebec, Canada, a comparison 4 ML models that estimated hourly SWT, showed SVR outperformed by RF (Hani et al., 2023).

A lesser-known form of SVM is its extended form, called *relevance vector machine (RVM)*. RVM is a form of supervised learning that uses a Bayesian framework to solve classification and regression problems (Tipping, 2001). *Locally weighted polynomials regression (LWPR)* is a form of supervised ML(Moore et al., 1997), used for learning continuous non-linear mappings from real-valued (i.e., functions whose values are real numbers) inputs and real-valued outputs. LWPR works by adapting the model locally to the respective data points, assigning different weights to different data points based on data point proximity to the target (Moore et al., 1997). This type of regression is best employed when the variance around the regression line is not constant, thereby suggesting heteroscedasticity.

**2.3.1.4 Gaussian Process Regression and Generalized Additive Models**

*Gaussian Process Regression (GPR)* is a type of nonparametric supervised learning methods used to solve regression problems. As a Bayesian approach, GPR assumes a probability distribution over all functions that fit the data. GPR is specified by a mean function and covariance kernel function which reflect prior knowledge of the trend and level of smoothness of the target function (Xu & Liang, 2021). One of GPR's advantages is the model's ability to calculate empirical confidence intervals, allowing the user to consider refitting predictions to areas of interest in the function space (Grbić et al., 2013). For more details on GPR, we refer the reader to Xu & Liang (2021). Grbic et al. (2013) used GPR for SWT modeling of the river Drava, Croatia, where model #1 estimated the seasonal component of SWT fluctuations and model #2 estimated the shorter-term component (Grbić et al., 2013). A separate study for the river Drava used three variations of GPR to model SWT, finding that GPR was outperformed by the physically based, stochastically calibrated model, air2stream (Zhu, Nyarko, Hadzima-Nyarko, Heddam, et al., 2019). More recently, Majerska et al. (2024) used GPR to simulate SWT for a non-glaciated arctic catchment, Fuglebekken (Spitsbergen, Svalbard). Using GPR and another model, the authors identified a diurnal warming trend of 0.5-3.5 °C per decade through the summer season, implying a warming thermal regime in the Fuglebekken catchment (Majerska et al., 2024).

Generalized Additive Models (GAMs) are a type of semi-parametric, non-linear model with a wide range of flexibility, allowing the model to analyze data without assuming relations between inputs and outputs (Hastie & Tibshirani, 1987). Where GPR uses a probabilistic approach, GAM uses smoothing functions, (i.e., splines) to model the relationship between a predictor variable and response variable. GAMs have been used to model SWT for the Sainte-Marguerite River in eastern Canada (Laanaya et al., 2017; Souaissi et al., 2023; Hani et al., 2023) with the latter-most study, Hani et al. (2023) using GAMs to identify potential thermal refuge areas for Atlantic salmon in two tributary confluences using sub-hourly observations.

**2.3.1.5 Decision trees and Classification and Regression Trees**

*Decision trees (DTs)* are a non-parametric, supervised learning technique. DTs can make predictions or decisions based on a set of input features and are likely to be more accurate where the problem can be solved in a hierarchical sequence of decisions (Breiman, 2001). *Classification and Regression Trees (CART)* is a specific type of algorithm that builds decision trees, where the internal node in the tree splits the data into two branches (sub-nodes) based on the specified decision rule (Loh, 2008). While CART can quickly find relationships between data, it is prone to overfitting and can be statistical

unstable, where a small perturbation in the training data could negatively affect the output of the tree (Hastie et al., 2001; Xu & Liang, 2021). For a detailed explanation of DT and CART, we refer the reader to Hastie et al. (2001), Loh (2008) and Xu and Liang (2021). A SWT modeling study comparing the output of three model versions of DT, GPR and feed-forward neural networks for multiple sites, found that DTs could perform similarly to GPR and feed-forward neural networks when detailed statistics of air temperature, day-of-year, and discharge were included (Zhu, Nyarko, Hadzima-Nyarko, Heddam, et al., 2019). However, when comparing daily SWT results from DTs with Gradient Boosting (GB) or Random Forest (RF), DTs generally underperform (Anmala & Turuganti, 2021; Jiang et al., 2022). Recent studies have compared CART with other ML algorithms to model water quality parameters (including SWT), finding that CART underperformed due to overfitting, compared to RF (Souaissi et al., 2023) and extreme learning machine, ELM (Heddam, Kim, et al., 2022). To combat the problem of overfitting that can occur using decision trees, the idea of using multiple trees by bootstrap aggregation (i.e., bagging), has gained interest.

### 2.3.1.6 Random Forests and XGBoost

RF and XGBoost, have been used to predict daily SWT prediction for Austrian catchments, with results showing minor differences in model performance, with a median RMSE difference of 0.08 °C between tested ML models (Feigl et al., 2021). Using RF and XGBoost along with four other ML models, Jiang et al. (2022) estimated daily SWT below dams in China, finding day of year, stream flow flux and AT to be most influential in the prediction of SWT (Jiang et al., 2022). Weierbach et al. (2022) used XGBoost and SVR to predict SWT at monthly time scales for the Pacific Northwest region of the U.S., showing that an ensemble XGBoost outperformed all modeling configurations for spatiotemporal predictions in unmonitored basins. with AT identified as the primary driver of monthly SWT. Zanoni et al. (2022) used RF and a deep learning model to develop regional models of SWT and other water quality parameters, with RF performance comparitively less effective at detecting non-linear relationships than the deep learning model, though both models identified AT as most influential (Zanoni et al., 2022).

Souaissi et al. (2023) tested the performance of RF and XGBoost, with non-parametric models for the regional estimation of maximum SWT at ungaged locations in Switzerland, finding no significant differences between the ML performance and the non-parametric model performances, which was attributed to the lack of a large dataset. Hani et al. (2023) used four supervised ML models - MARS, GAM, SVM, and RF to model potential thermal refuge area (PTRA) at an hourly timestep for two tributary confluences of the Sainte-Marguerite River in Canada. RF had the highest accuracy at both locations in terms of hourly PTRA estimates and modeling SWT (Hani et al., 2023). Wade et al. (2023) conducted a CONUS-scale study using 410 USGS sites with four years of daily SWT and discharge to examine maximum SWT. They used RF to estimate max SWT and thermal sensitivity (Wade et al., 2023), finding that AT was the most influential control followed by other properties (watershed characteristics, hydrology, anthropogenic impact).

### 2.3.2 Traditional artificial neural networks (ANN)

An *artificial neural network (ANN)* is a type of ML algorithm inspired by biological neural networks in the brain (McCulloch & Pitts, 1943; Hinton, 1992). ANNs learn from data provided and improve on their own to progressively extricate higher-level trends or relationships within the given dataset (Hinton, 1992). Currently, ANNs are capable of data classification, pattern recognition, and regression analysis. Considered robust, ANNs can undergo supervised, unsupervised, semi-supervised and reinforcement learning. The first study that utilized ANNs specifically for SWT modeling was published around the year 2000. The work was done by researchers interested in hindcasting SWT for a river in Canada, for a 41-year period dating back to 1953 (Foreman et al., 2000). Since 2000, various types of ANNs have been increasingly used to model SWT at various sites at hourly, daily and monthly time steps. For more detail on traditional ANNs, with descriptions of ANN variants and backpropagation alternatives, we refer the reader to Appendix A.

### 2.3.3 Newer/recent ML algorithms

We define newer/recent ML algorithms as those introduced or re-introduced in the last decade for SWT modeling. These ML algorithms include deep (i.e., increased layers) ANNs such as recurrent neural networks (RNN), convolutional neural networks (CNN), extreme learning machine (ELM), ML hybridizations and subsets.

A *"deep" neural network (DNN)* has three or more hidden layers, MLPNNs being one such example. The purpose of added layers is to serve as optimizations for greater accuracy. Due to their complex nature, DNNs need extensive time spent solely on training the network on the input data (Abdi et al., 2021). *Convolutional neural networks (CNN)* are FFNNs used to recognize objects and patterns in visual data (LeCun et al., 1989, 2004). CNNs have convolutional layers, which hold one or more filters that calculate a local weighted sum as they analyse the input data. A CNN filter is a matrix (rows and columns) of randomized number values that convolves (i.e., moves), through the pixels of an image, taking the dot product of the matrix of values in the filter and the pixel values of the image. The dot product is used as input for the next convolutional layer. To ensure adequate performance, CNNs must be trained with examples of correct output in the form of labelled training data and should be calibrated (i.e., adjusting filters, implement loss functions) to optimize performance (Krizhevsky et al., 2012). For more detail on CNN, we refer the reader to LeCun et al. (2004), Krizhevsky et al., (2012) and Xu and Liang (2021). A disadvantage of CNNs is that they are not ideal for interpreting temporal or sequential information or data that require learning from past data to predict future output. For interpreting temporal information or sequential data, recurrent neural networks are preferred.

Unlike FFNNs, recurrent neural networks (RNN) work in a chain-link nature that allows them to loop (i.e., keep) previously handled data for use in a present task to make better predictions (Hochreiter & Schmidhuber, 1997). The RNN architecture is better equipped (and preferred) to handle temporal (i.e., time series) or sequential (i.e., a video is a sequence of images) data due to their ability to learn from their past (Bengio et al., 1994). The *Elman neural network (ELM-NN)* is a

type of RNN where the hidden layer (bi-directionally connected to the input layer and output layer) stores contextual information of the input that it sends back to the input layer with sequential time steps (Elman, 1990).

However, one of the issues that persists in RNNs, is that there is a limit to how far back RNNs can access past data to make better predictions. This is described as the problem of long-term dependencies, also known as the vanishing gradient problem. The vanishing gradient problem is due to backpropagated gradients that can grow or shrink at each time step, increasing instability until the gradients "explode" or "vanish" (Bengio et al., 1994; Hochreiter & Schmidhuber, 1997). Hochreiter & Schmidhuber (1997) introduced the *long short-term memory (LSTM)* model, a type of RNN explicitly designed

to overcome the vanishing gradient problem. The LSTM architecture includes three gates (input, forget, and output gates) that control the flow of information in and out of the cell state, allowing the ANN to store and access data over longer time periods. In the last few decades, LSTMs have improved, and variations introduced (Gers & Schmidhuber, 2000; Cho et al., 2014; Yao et al., 2015) and many have been cross compared, with findings showing similar performance across LSTMs (Greff et al., 2016). In the last few years, LSTMs and their variations have been revisited and employed in hydrologic studies

to examine possible relationships in time series data (Shi et al., 2015; Shen, 2018; Kratzert et al., 2018, 2019). For example, Sadler et al. (2022) used a LSTM model to multi-task, i.e., predict two related variables - streamflow and SWT. Their argument for forcing a LSTM to multi-task is that if two variables are driven by the same underlying physical processes, a multi-tasking LSTM could more wholistically represent shared hydrologic processes and thus better predict the variable of interest. Their LSTM model consisted of added components: specifically, two parallel, connected output layers that

represented streamflow output and SWT output (Sadler et al., 2022). Overall, using the multi-tasking LSTM improved accuracy for half the sites, but for those sites with marked improvement, more calibration was needed to reach improvement (Sadler et al., 2022).

    Another type of NN is the *graph neural network (GNN),* which is used for representation learning (unsupervised learning of feature patterns) of graphed data, where a "graph" denotes the links between a collection of nodes. At each graph

node or link, information in the form of scalars or embeddings can be stored, making them very flexible data structures. Example of graphs that we interact with regularly are images, where each pixel is a node and is linked to adjacent pixels. A stream network is also an example of a graph, albeit a directed graph, which is a graph in which the links (also called 'edges') have direction. Two examples of recent GNNs are *recurrent graph convolution networks (RGCN)* and *temporal convolution graph models (TCGM)*. The RGCN utilizes LSTM network architecture (i.e., use of forget, input, output gates)

for temporal recognition (Topp et al., 2023). In contrast to RGCN, TCGM uses 1D convolutions (i.e., input a 3-dimensional object and output a 3-dimensional object), pooling, and channel-wise normalization to capture low-, intermediate- and high-level temporal information in a hierarchical manner (Lea et al., 2016). An example that utilizes this approach is Graph Wave Net (Wu et al., 2019), which has been used in spatial-temporal modeling of SWT (Topp et al., 2023). According to Topp et al. (2023), the temporal convolutional structure of Graph Wave Net is more stable in the gradient-based optimization process

in contrast to the possible gradient explosion problem that the LSTM in the RGCN could experience.

While present studies continue to use ML models as standalones to evaluate SWT predictions, other studies have coupled modern ML with non-ML models to examine whether such combinations improve model performance (Graf et al., 2019; Qiu et al., 2020; Rehana & Rajesh, 2023). For example, Graf et al. (2019) coupled four discrete wavelet transform (WT) techniques with MLPNN to predict SWT for eight stations on the Warta River in Poland. For reference, WT is widely applied for the analysis and denoising of information (signals) and images both over time and on a domain scale (frequency). The unique characteristic of a wavelet neural network (WNN) is the use of the WT as the activation function in the hidden layer of the NN (Qiu et al., 2020). Zhu, Hadzima-Nyarko, Gao et al. (2019) coupled WT with MLPNN and ANFIS to evaluate daily SWT at two stations on the River Drava in Croatia and separately compared the WT-ML coupling with MLR. The study found that the combination of WT and ML improved performance compared to the standalone models (Zhu, Hadzima-Nyarko, Gao, Wang, et al., 2019).

A recent ML approach called differentiable modeling, incorporates physics into ML modeling frameworks, where the basic model structure and parameters of a process-based model are inserted into an ANN to estimate parameters or replace existing process descriptions (Rahmani et al., 2023). Rahmani et al. (2023) examined model components that could improve a LSTM model's ability to better match model predictions to field observations. From their study, Rahmani et al. (2023) found that adding a separate shallow subsurface flow component to the LSTM model, and a recency-weighted averaging of past air temperature for calculating source SWT resulted in improved predictions (Rahmani et al., 2023).

Attention-based transformers are a more novel type of deep learning that has led to advancements in natural language processing, in the form of ChatGPT, Microsoft's CoPilot, Google's Gemini and others. Due to their exponential success in the last few years, attention-based transformer models have been used in geological science fields such as oceanography for sea surface temperature prediction (Shi et al., 2024), hydrology for streamflow and runoff prediction (Ghobadi and Kang, 2022; Wei, 2023) and remote sensing for streambed land use change classification (Bansal and Tripathi, 2024). As a relatively new AI tool, attention-based transformers have yet to be used for SWT (to our knowledge), but their applications in other geological science fields suggest it is only a matter of time before their use is observed in SWT modeling.

## 2.4 SWT predictions using ML

### 2.4.1 Identifying Model Complexity

The strong success of ML-use in SWT modeling warrants a brief and broad overview on identifying model complexity to minimize overfitting and underfitting of models. When a model is too complex, i.e., has too many features or parameters relative to the number of observations, or is forced to overextend its capabilities, i.e., make predictions with insufficient training data, the model runs the risk of overfitting (Srivastava et al., 2014). An overfitted model fits the training data "too well", capturing noise and details that provide high accuracy on a training dataset, only to perform poorly once the model encounters "unseen" data in testing/validation (Xu and Liang, 2021). Scenarios where overfitting may be temporarily acceptable are: 1) model development is at preliminary stages, such as a "proof of life" concept, 2) when the objective is to

identify heavily relied on features by the model, i.e., feature importance, or 3) in highly controlled modeling environments where the expected data will be consistently similar to the training dataset. The latter is more likely in industrial applications and unlikely in the changing nature of hydrology.

In contrast, underfitting occurs when a model is too simple to capture any patterns in the data, which can also lead to unsatisfactory performance in training, testing and validation. Underfitting can occur with inadequate model features, poor model complexity or when regularization techniques, (e.g., L1 or L2 regularization), are over-used, making the model too rigid and unable to respond to changes in the data. Given the propensity of ML models to effectively learn the training data, underfitting is less an issue in ML whereas overfitting can be widespread. In Figure 1, we present an example workflow that researchers can use to transition away from overfitting and towards model generalizability. In the five-step outline (Fig. 1), we suggest the need for "Temporal, Unseen, Ungaged Region Tests" (TUURTs) in SWT ML modeling. The idea behind TUURTs has been applied for decades in SWT process-based (Dugdale et al., 2017) and statistically based models (Benyahya et al., 2007; Gallice et al., 2015) to improve SWT model robustness. In TUURTs, testing for "unseen" cases means testing only within the developmental dataset, whereas testing for "ungaged" cases means testing for new sites that have no data and have not been previously seen by the model at all. Some statistically based models, such as DynWat (Wanders et al., 2019) and the Pacific Northwest (PNW) SWT model (Siegel et al., 2023) have tested for ungaged regions and unseen data. In the last few years, ML-SWT studies have begun applying TUURTs (Rahmani et al., 2020, 2021, 2023; Topp et al., 2023; Hani et al., 2023, Souassi et al., 2023; Philippus, Sytsma, et al., 2024) but more ML-SWT studies need to apply these tests to improve user confidence in extrapolation capability. We further encourage researchers to shift towards more generalizable models, which are in theory, more capable of performing well across diverse scenarios and datasets and stand to become increasingly important with the unpredictability of climate extremes.

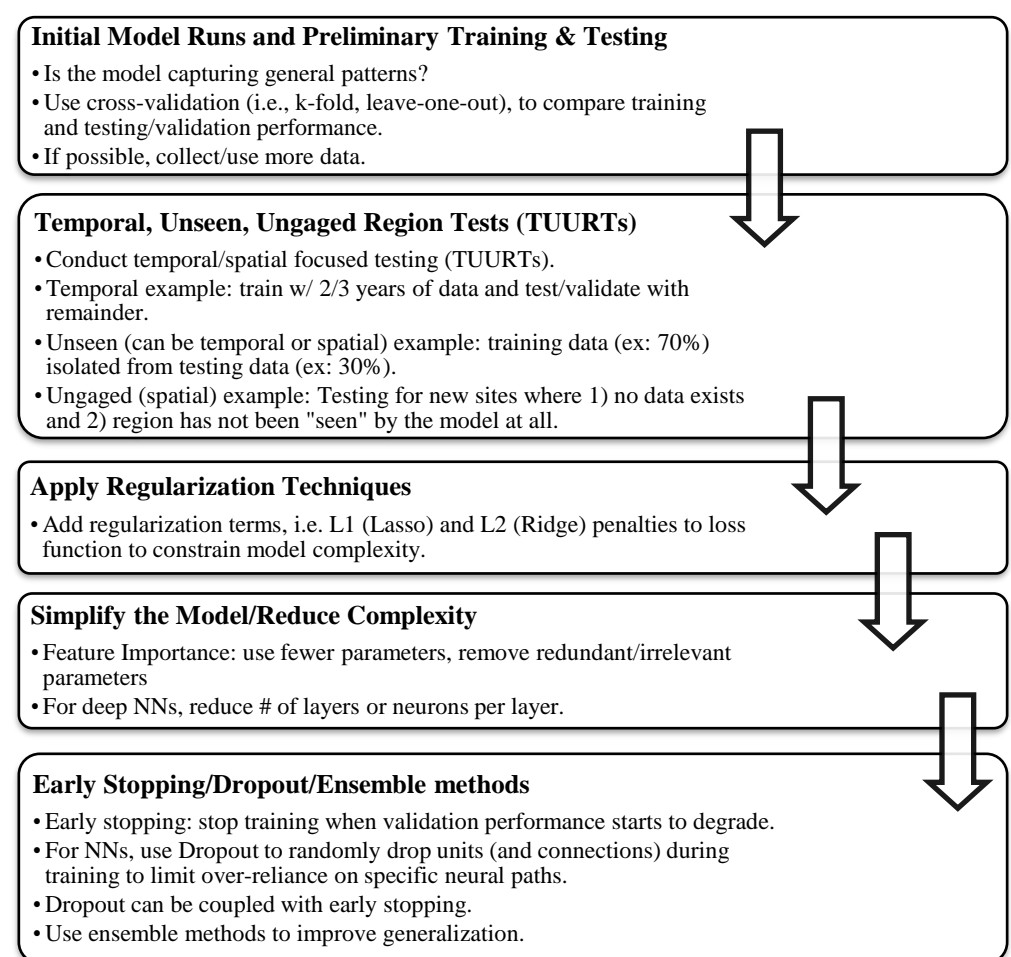

**Initial Model Runs and Preliminary Training & Testing**
• Is the model capturing general patterns?
• Use cross-validation (i.e., k-fold, leave-one-out), to compare training and testing/validation performance.
• If possible, collect/use more data.

**Temporal, Unseen, Ungaged Region Tests (TUURTs)**
• Conduct temporal/spatial focused testing (TUURTs).
• Temporal example: train w/ 2/3 years of data and test/validate with remainder.
• Unseen (can be temporal or spatial) example: training data (ex: 70%) isolated from testing data (ex: 30%).
• Ungaged (spatial) example: Testing for new sites where 1) no data exists and 2) region has not been "seen" by the model at all.

**Apply Regularization Techniques**
• Add regularization terms, i.e. L1 (Lasso) and L2 (Ridge) penalties to loss function to constrain model complexity.

**Simplify the Model/Reduce Complexity**
• Feature Importance: use fewer parameters, remove redundant/irrelevant parameters
• For deep NNs, reduce # of layers or neurons per layer.

**Early Stopping/Dropout/Ensemble methods**
• Early stopping: stop training when validation performance starts to degrade.
• For NNs, use Dropout to randomly drop units (and connections) during training to limit over-reliance on specific neural paths.
• Dropout can be coupled with early stopping.
• Use ensemble methods to improve generalization.

Figure 1. Diagram outlining steps that can be taken in modeling process to mitigate overfitting.

### 2.4.2 Model Inputs for ML-SWT

Using air temperature (AT) to better understand SWT has been considered since the 1960s, when Ward (1963) and Edinger et al. (1968) discussed the influence of air temperature on SWT. Since then, various input variables have been tested (see Table S1), however, the model inputs of AT and SWT continue to be the most used in ML-modeling studies. For example,

studies have used AT from time periods outside of the known SWT record to improve ML model performance (Sahoo et al., 2009; Piotrowski et al., 2015; Graf et al., 2019). In addition to AT and SWT, flow discharge has been used to attempt to constrain SWT (Foreman et al., 2001; Tao et al., 2008; St-Hilaire et al., 2011; Grbić et al., 2013; Piotrowski et al., 2015; Graf et al., 2019; Qiu et al., 2020). Other model inputs include precipitation (Cole et al., 2014; Jeong et al., 2016; Rozos, 2023), wind direction/speed (Hong and Bhamidimarri, 2012; Cole et al., 2014; Jeong et al., 2016; Kwak et al., 2016;

Temizyurek and Dadaser-Celik, 2018; Abdi et al., 2021; Jiang et al., 2022), barometric pressure (Cole et al., 2014), landform

attributes (Risley et al., 2003; DeWeber and Wagner, 2014; Topp et al., 2023; Souaissi et al., 2023), and many more (see Table S1).

In the last few years, including the day-of-year as an input, DOY (Qiu et al., 2020; Heddam et al., 2022; Drainas et al., 2023; Rahmani et al., 2023) and humidity (Cole et al., 2014; Hong and Bhamidimarri, 2012; Kwak et al., 2016; Temizyurek and Dadaser-Celik, 2018; Abdi et al., 2021), have also shown to better capture the seasonal patterns of SWT (Qiu et al., 2020; Philippus, Sytsma, et al., 2024). With improved access to remote sensing data, there has also been a notable increase of satellite product inputs such as estimates of sky cover (Cole et al., 2014), solar radiation (Kwak et al., 2016; Topp et al., 2023; Majerska et al., 2024), sunshine per day (Drainas et al., 2023) and potential ET (Rozos, 2023; Topp et al., 2023). However, more research is needed to better understand the influence of newer model inputs on SWT (Zhu and Piotrowski, 2020).

Recently, SWT studies focused on the CONUS-scale have chosen to use as many model inputs as available, with Wade et al. (2023), a point-scale CONUS ML study using over 20 variables, while Rahmani et al. (2023) created a LSTM model and considered over 30 variables to simulate SWT. Despite the use of diverse data, the models in these studies performed only satisfactorily and were deemed not generalizable, leaving much room for improvement in CONUS-scale modeling of SWT. With the compilation of larger and larger datasets, feature importance in ML, that is the process of using techniques to assign a score to model input features based on how good the features are at predicting a target variable, can be an efficient way to improve data comprehension, model performance, and model interpretability, the latter of which can dually serve as a transparency marker of which features are driving predictions. Methods for measuring feature importance include using correlation criteria (Pearson's $r$, Spearman's $rho$), permutation feature importance (shuffling feature values, measuring decrease in model performance), linear regression feature importance (larger absolute values indicate greater importance), or if using CART/RF/gradient boosting, entropy impurity measurements can be insightful (Venkateswarlu and Anmala, 2023).

For example, one technique that can be used to improve ML model parameter selection is the Least Absolute Shrinkage and Selection Operator (LASSO), a regression technique used for feature selection (Tibshirani, 1996). Research utilizing ML models for SWT frequency analysis at ungaged basins used the LASSO method to select explanatory variables for two ML models (Souaissi et al., 2023). The LASSO method consists of a shrinkage process where the method penalizes coefficients of regression variables by minimizing them to zero (Tibshirani, 1996). The number of coefficients set to zero depends on the adjustment parameter, which controls the severity of the penalty. Thus, the method can perform both feature selection and parameter estimation, an advantage when examining large datasets (Xu & Liang, 2021).

### 2.4.3 Local: Single rivers, Site-specific ($\leq 100$ km$^2$)

SWT predictions using ML have extended from the local scale to nearly continental scales over the last 24 years. One of the first studies to use a neural network to estimate SWT used a MLPNN was done by Sivri et al., (2007) who predicted monthly SWT for Firtina Creek in Türkiye, a novel approach at the time. While the MLPNN model $R^2 \sim 0.78$ was not very good, the proof of concept was a success (Sivri et al., 2007). Chenard and Caissie (2008) used eight ANNs to calculate daily and max

SWT for Catamaran Brook, a small drainage basin tributary to the Miramichi River in New Brunswick, Canada for the years 1992 to 1999. Their ANN models performed best in late summer and autumn and performed comparatively to stochastic models for the same watershed (Chenard & Caissie, 2008). In 2009, Sahoo et al. (2009) compared an ANN, multiple regression analysis, and dynamic non-linear chaotic algorithms (Islam & Sivakumar, 2002) to estimate SWT in the Lake Tahoe watershed area in along the California/Nevada border within the U.S. Their ANN models included available solar radiation and air temperature, with results showing a variation of the BPNN as having the best performance (Sahoo et al., 2009).

Hadzima-Nyarko et al. (2014) used a linear regression model, a stochastic model, and variations of two NNs: MLP (six variations) and RBF (two variations), to compute and compare SWT predictions for four stations on the river Drava, along the Croatia-Hungary border in southern Central Europe. While their ANN models performed better than the linear regression and stochastic models, a comparison of their NN models found that one of their six MLPNN variations barely outperformed the RBFNN, with a difference in RMSE of 0.0126 °C, within the margin of error. The authors stated that apart from the current mean AT, the daily mean AT of the prior two days and classification of the day of the year (DOY) were significant controls of the daily SWT (Hadzima-Nyarko et al., 2014). Rabi et al. (2015) conducted a study using the same gage stations on the river Drava using only AT as a predictor and restricted the use of NNs to only MLP, finding that the MLPNN outperformed the linear regression approaches (Rabi et al., 2015).

Cole et al. (2014) tested a suite of models including an FFNN to predict SWT downstream of two reservoirs in the Upper Delaware River, in Delaware, U.S. During training, the FFNN was outperformed by an Auto Regressive Integrated Moving Average (ARIMA) model and performed similarly to the physically based Heat Flux Model, HFM (Cole et al., 2014). During testing, the FFNN, ARIMA, and HFM models performed similarly, with HFM being slightly more accurate due to its advantage as a physically based model with data availability and calibration potential (Cole et al., 2014). The authors suggest that the under/over-predictions of the models may have been from unaddressed groundwater inputs or unaccounted for nonlinear relationships (Cole et al., 2014). Hebert et al. (2014) focused on the Catamaran Brook area (like Chenard and Caissie, 2008) and included the Little Southwest Miramichi River in New Brunswick, Canada, to conduct ANN model predictions of hourly SWT. The study considered spring through autumn, hourly data from 1998 to 2007, finding that the ANN models performed similarly or better than deterministic and stochastic models for both areas (Hebert et al., 2014).

Piotrowski et al. (2015) examined data from two streams, one mountainous and one lowland, in a moderately cold climate of eastern Poland, to model SWT using MLPNN, PUNN, ANFIS, and WNN. The ANN models were independently calibrated to find the best fits, with results showing that MLPNN and PUNN slightly outperformed ANFIS and WNN (Piotrowski et al., 2015). The study also found current AT, and information on the mean, maximum, and minimum AT from 1-2 days prior as important for improving model accuracy (Piotrowski et al., 2015). Temizyurek and Dadaser-Celik (2018) used an ANN with observations of AT, relative humidity, prior month SWT and wind speed to predict monthly SWT at four gages on the Kızılırmak River in Turkey. Best results were obtained from using the sigmoidal (S-shape) activation function

and the scaled conjugate gradient algorithm (Møller, 1993), though the average RMSE (~2.3 °C) for the NN used was higher (worse) than the average calculated from this literature review where RMSE ~1.4° C.

Zhu et al. (2019) conducted four studies that used NNs to examine SWT on the river Drava, Croatia (Zhu, Hadzima-Nyarko, Gao, Wang, et al., 2019; Zhu, Heddam, Nyarko, et al., 2019; Zhu, Nyarko, Hadzima-Nyarko, Heddam, et al., 2019). They also examined SWT of three rivers in Switzerland (Zhu, Heddam, Nyarko, et al., 2019; Zhu, Nyarko, Hadzima-Nyarko, Heddam, et al., 2019), and three rivers in the U.S. (Zhu, Heddam, Wu, et al., 2019; Zhu, Nyarko, Hadzima-Nyarko, Heddam, et al., 2019). Across the studies, the MLPNN models had better performance compared to ANFIS (Zhu, Heddam, Nyarko, et al., 2019), GPR (Zhu, Nyarko, Hadzima-Nyarko, Heddam, et al., 2019), or MLR (Zhu, Heddam, Wu, et al., 2019). Qiu et al. (2020) used variations of NNs (MLP/BPNN, RBFNN, WNN, GRNN, ELMNN) to examine SWT at two stations on the Yangtze River, China, finding that the MLP/BPNN outperformed all other models when the particle swarm algorithm (PSO) was used for optimization (Qiu et al., 2020). Stream discharge and DOY were also shown to improve model accuracy. Piotrowski et al. (2020) used various MLPNN shallow (one hidden layer) structures to test the use of an approach called dropout in SWT modeling using data from six stations in Poland, Switzerland, and the U.S. The dropout approach can be applied to deep ANNs due to its efficiency in preventing overfitting and low computation requirements (Piotrowski et al., 2020). The study found that use of dropout and drop-connect significantly improved performance of the worst training cases. For more information on the use of dropout with shallow ANNs, we refer the reader to Piotrowski et al. (2020).

Graf and Aghelpour (2021) compared stochastic and ANN (ANFIS, RBF, GMDH) SWT models for four gages on the Warta River, in Poland, finding that all models performed similarly well ($R^2$ > 97.6 %). Results showed that the stochastic and ML models performed similarly, while the stochastic models had less prediction errors for extreme SWT (Graf & Aghelpour, 2021). Rajesh and Rehana (2021) used several ML models (Ridge regression, K-nn, RF, SVR) to predict SWT at daily, monthly and seasonal scales for a tropical river system of India. The authors found that the monthly SWT prediction performed better than the daily or seasonal (Rajesh & Rehana, 2021). Of the ML models, the SVR was the most robust, though a data assimilation algorithm notable improved predictions (Rajesh & Rehana, 2021). Jiang et al. (2022) examined SWT under the effects of the Jinsha River cascaded reservoirs using six ML models (i.e., adaptive boosting, AB, decision tree, DT, random forest, RF, support vector regression, SVR, gradient boosting, GB, and multilayer perceptron neural network, MLPNN). The study found that day of year (DOY) was most influential in each model for SWT prediction, followed by stream flow and AT (Jiang et al., 2022). With knowledge of the influential parameters, ML model variations were tested, finding that gradient boosting and random forest provided the most accurate estimation for the training dataset and the test dataset (Jiang et al., 2022). Abdi et al. (2021) used linear regression and a deep (multi-layered) neural network (DNN) to predict hourly SWT for the Los Angeles River, finding that the DNN outperformed the linear regressions. They suggested that using a variety of ML models to predict SWT could add robustness to a study, but state that training ANNs is more time-consuming than training linear regression models, for minimal improved accuracy (Abdi et al., 2021).

Khosravi et al. (2023) used an Exploratory Data Analysis (EDA) technique, a type of feature engineering that prepares the dataset for best performance with an LSTM to identify SWT predictors (discharge, water level, AT, etc.) up to one week

in advance for a monitoring station on the Central Delaware River. The authors noted that though the LSTM performed satisfactorily, future studies should compare LSTMs with CNNs or other model types, and that generalizability is limited to the specific location and dataset (Khosravi et al., 2023). Majerska et al. (2024) used GPR to simulate SWT for the years 2005-2022 for the artic catchment Fuglebekken in Svalbard, Norway. The unique opportunity to study SWT of an unglaciated High Arctic stream regime showed an alarming warming throughout the summer where SWT increased as much as 6 °C, highlighting a strong sensitivity of the Arctic system to ongoing climate change (Majerska et al., 2024).

### 2.4.4 Regional, Continental Scale (≥ 100 km$^2$)

DeWeber and Wagner (2014) conducted one of the first regional ANN ensemble studies, focusing on thousands of individual streams reaches across the Eastern U.S. They used an ensemble of 100 ANNs to estimate daily SWT with varying predictors for the 1980 - 2009 period, finding that daily AT, prior 7-day mean AT and catchment area were the most important predictors (DeWeber & Wagner, 2014). In Serbia, Voza and Vukovic (2018) conducted cluster analysis, PCA and discriminant analysis for the Morava River Basin using data from 14 river stations to identify monitoring periods for sampling. With discriminant parameters identified, an MLPNN was used to predict changes in the values of the discriminant factors (see fig. 1 of Voza and Vukovic, 2018) and identify controls on the monitoring periods, finding that seasonality and geophysical characteristics were most influential (Voza & Vuković, 2018).

Rahmani et al. (2020) used four years of SWT data for 118 sites across the CONUS to test three LSTM models that simulated SWT, finding that the LSTM trained with streamflow observations was the most accurate, which was unsurprising. Of interest to the reader would be the inner mechanisms of the LSTM, but the study did not explicitly state what physical laws were followed by the LSTM. Instead, the authors hypothesized that the LSTM could assume internal representations of physical quantities (i.e., water depth, snowmelt, net heat flux, baseflow temperature, SWT). The authors further stated that the LSTM was dependent on a good historical data record and would not generalize well to ungaged basins. A follow-up study by Rahmani et al. (2021) used six years of SWT data and relevant meteorological parameters for 455 sites across the CONUS (minus California and Florida) to test LSTM models for data-scarce, dammed, and semi-ungaged basins (discharge used as input). The follow-up study showed improved performance, but the models remained limited in capturing the influence of latent contributions such as base-flow and subsurface storage. Feigl et al. (2021) tested the performance of six ML models: stepwise linear regression, RF, XGBoost, FFNNs, and two RNNs (LSTM and GRU) using data from 10 gages in the Austria-Germany-Switzerland region, to estimate daily SWT. From the comparison, FFNNs and XGBoost were the best performing in 8 of 10 catchments (Feigl et al., 2021). For modeling SWT in large catchments (> 96,000 km$^2$ ~ Danube catchment size), the RNNs performed best due to their long-term dependencies (Feigl et al., 2021). Zanoni et al. (2022) used RF, DNN, and a linear regression to predict daily SWT in the Warta River basin and compared the results with those of stochastic models. Their results found that the DNN was the most effective in capturing nonlinear relationships between drivers (i.e., SWT) and water quality parameters (Zanoni et al., 2022). On parameter influence, the

analysis also found that DOY was an adequate surrogate for AT input in modeling SWT, experiencing only a slight performance reduction.

Heddam, Ptak, et al., (2022) used six ML models: K-nn, LSSVM, GRNN, CCNN, RVM, and LWPR, to evaluate SWT for several of Poland's larger rivers. For each ML, three variations were created: one calibrating with only AT as input, another calibrating with AT and DOY, and a third decomposing AT using the variational mode decomposition, VMD (Heddam, Ptak, et al., 2022). For more on VMD, we refer the reader to Heddam, Ptak, et al. (2022). The study found that the VMD parameters improved RMSE and MAE performance metrics for some models, but neither GRNN nor *K-nn* showed

improvement. Heddam, Kim, et al., (2022) examined how use of the Bat algorithm optimized the extreme learning machine (Bat-ELM) neural network and how that in turn affected modeling of SWT in the Orda River in Poland. Results from the Bat-ELM were compared with MLPNN, CART, and multiple linear regression, MLR, finding the Bat-ELM outperformed MLPNN, CART and MLR (Heddam, Kim, et al., 2022). Focusing on a region of Germany, Drainas et al. (2023) trained and tested various ANNs with different inputs, for 16 small ($\leq 1$ $m^3$ $s^{-1}$) headwater streams, finding that the best performing

(lowest RMSE) input combination was stream-specific, suggesting that the optimal input combination cannot be generalized across streams for the region (Drainas et al., 2023). The ANN prediction accuracy of SWT was negatively affected by river length, total catchment area and stream water level (Drainas et al., 2023). Additionally, ANN accuracy suffered when dealing with open-canopy land use types such as grasslands but improved with semi-natural and forested land cover (Drainas et al., 2023). Recently, He et al. (2024) built a LSTM framework to model water dynamics in stream segments while

attempting to capture spatial and temporal dependencies. First, a baseline LSTM+GNN, then improved it by using graph masking and adjusting the model based on constraints (He et al., 2024). For the Delaware River Basin, the Fair-Graph model performed slightly better than the baseline with a RMSE of 1.83 vs. 1.78, respectively. For the Houston River network, the Fair-Graph model also performed slightly better than the baseline (NSE of 0.721 vs. 0.580). While the relative performance compared to baseline was not significantly better, we anticipate that graph masking (algorithm that incorporates spatial

awareness into ANN), will play an increasingly large role in hydrologic modeling (Shen, 2018; He et al., 2024).

**2.5. Decision Support and Climate Change Scenarios**

In 2003, the United States Geological Survey (USGS), used a FFNN to estimate hourly SWT for a summer season in western Oregon (Risley et al., 2003). Their work used the predicted SWT to better constrain future total maximum daily loads (TMDL) for stream management. Jeong et al. (2016) used an ANN to evaluate SWT for the Soyang River, South Korea. The

goal was to couple the ANN predictions with a cyber infrastructure prototype system to deliver automated, real-time predictions using weather forecast data (Jeong et al., 2016).

Liu et al. (2018) used a hydrological model called the Variable Infiltration Capacity (VIC) model to produce estimates of AT and river section-based variables for the Eel River Basin, Oregon, U.S., to be used as input data for an ANN. The study considered the AT rise from the RCP 8.5 scenario to estimate future (2093-2100) daily stream flow and SWT, finding

that SWT was increasingly sensitive to the proportion of base flow in the summer (Liu et al., 2018). Topp et al. (2023) used

the Delaware River Basin in the eastern U.S. to compare two DL models: a recurrent graph convolution network (RGCN), and a temporal convolution graph model, TCGM, called Graph Wave Net. The comparison included scenarios capturing climate shifts representative of long-term projections where warm conditions or drought persisted. Considered spatiotemporally aware, the two process-guided deep learning models performed well (test RMSE of 1.64 °C and 1.65 °C); however, Graph WaveNet significantly outperformed RGCN in four of five experiments where test partitions represented diverse types of unobserved environmental conditions.

Further focusing on the Delaware River Basin, Zwart et al. (2023a) used data assimilation and an LSTM to generate 1-day and 7-day forecasts of daily maximum SWT for the purpose of aiding reservoir managers in decisions about when to release water to cool streams. Following up on this study was Zwart et al. (2023b), who used a LSTM and a RGCN, both with and without data assimilation, to generate 7-day forecasts of daily maximum SWT for monitored and unmonitored locations in the Delaware River Basin, finding that the RGCN with data assimilation performed best for ungaged locations and at higher SWT, which is important for reservoir operators to be aware of while drafting release schedules.

Rehana and Rajesh (2023) used a standalone LSTM, a WT-LSTM, and a k-nearest neighbour (K-nn) bootstrap resampling algorithm with LSTM, to assess climate change impacts on SWT using downscaled projections of AT with RCPs of 4.5 and 8.5 for seven polluted river catchments in India. Comparing the coupled models and the physically based air2stream model, they found the K-nn coupled with LSTM to be the best performing in terms of effectively predicting SWT at the monthly time scale. Considering the RCP scenarios, the predicted SWT increase for 2071-2100 for the rivers in India, ranged from 3.0-4.7 °C.

## 3   Model Evaluation Metrics

The second part of this review was to compile ML performance evaluation metrics as it pertains to SWT modeling and prediction and consider the commonly used metrics and suggest guidelines for easier comparison of ML performance across SWT studies. We considered journal articles from 2000-2024 that used ML to evaluate, predict, or forecast SWT and examine what model performance metrics had been used. Performance metrics can be calculated during model calibration, testing, and (or) validation to compute a single value that denotes the agreeableness between simulated and observed data.

For this literature review, all journals examined used at least one metric to evaluate model performance, with two or more metrics used by > 84 % of studies published on or after the year 2019. For review, the quantitative statistics were split into three categories: standard regression, dimensionless and error index (Moriasi et al., 2007). Standard regression statistics (Pearson's $r$, $R^2$) are ideal for examining the strength of the linear relationship between model simulations/predictions and the observed/measured data. Dimensionless techniques (NSE, KGE) provide a relative assessment of model performance, but due to their interpretational difficulty (Legates & McCabe, 1999) have been less commonly used. In contrast, error indices (RMSE, MAE) quantify the error in terms of the units of the data (i.e., °C) considered.

### 3.1 Model Performance Metrics: Standard Regression

The most basic statistics (slope, y-intercept mean, median, standard deviation) continue to be use in part due to their simplicity and ease of interpretability. These statistics are useful for preliminary examinations, where the assumption is that measured and simulated values are linearly related, and all the variance of error is contained within the predictions/simulations, whilst the observations are free of error. Unfortunately, observations are rarely error-free, and datasets are nonlinear, highlighting a need for using a diverse set of statistics (Helsel & Hirsch, 2002). One such set of statistics that are commonly used for standard regressions are called the correlation coefficients - Kendall's tau, Spearman's rho, and Pearson's $r$.

Pearson's $r$, also known as the correlation coefficient, is used to determine the strength and direction (i.e., positive, negative) of a simple linear relationship (Helsel & Hirsch, 2002). Values of $r$, range from -1 to +1, where $r < 0$ indicates a negative correlation and $r > 0$ indicates a positive correlation (Legates & McCabe, 1999). The square of $r$ is denoted as $r^2$, known as the square of the correlation coefficient, with values of $r^2$ ranging from 0 to 1. The $r^2$ metric is commonly used in simple linear regression to assess the goodness of fit by measuring the fraction of the variance in one variable (i.e., observations) that can be explained by the other variable (i.e., predictors). The metric $r^2$ tends to be confused with $R^2$, the latter which is a statistical measure that represents the proportion of variance explained by the independent variable(s) in a multiple linear regression model (Helsel and Hirsch, 2002). Part of the confusion may be related to the fact that $R^2$ shares the same range of 0 to 1, with $R^2 = 1$ indicating that the model can explain all the variance, and vice versa. We note that while both $r^2$ and $R^2$ share similarities in that they measure the proportion of variance, $R^2$ is more commonly used for multiple linear regression context, while $r^2$ is best suited for simple linear regressions. To reduce confusion, we strongly suggest that $r$, $r^2$ and $R^2$ always be reported together (even if as a supplement to a manuscript) to characterize goodness-of-fit.

In contrast to the linear regression metrics,, Spearman's rank correlation coefficient, rho ($\rho$) is a non-parametric rank-sum test useful for analysing non-normally distributed data and nonlinear monotonic relationships (Helsel & Hirsch, 2002). The data is ranked on a range from -1 to +1, where $\rho = 0$ indicates no association and $\rho$ = -1 or +1 suggest a perfect monotonic relationship. By ranking the data, Spearman's correlation coefficient quantifies monotonic relationships between two variables (converts nonlinear monotonic relationships to linear relationships) allowing $\rho$ to be robust against outliers (Helsel & Hirsch, 2002).

### 3.2 Model Performance Metrics: Error Indices

The mean absolute error (MAE), mean square error (MSE) and root mean squared error (RMSE) are popular error indices used to assess model performance. The equations for MAE, MSE, and RMSE are:

$$MAE = \frac{1}{N} \sum_{i=1}^{N} |O_i - P_i| \qquad \text{eq. 1}$$

$$MSE = \frac{1}{N}\sum_{i=1}^{N}(P_i - O_i)^2 \qquad \text{eq. 2}$$

$$RMSE = \sqrt{\frac{\sum_{i=1}^{N}(P_i - O_i)^2}{N}} \qquad \text{eq. 3}$$

For the equations, $N$ is the number of samples, $O_i$ is the observed SWT and $P_i$ is the predicted SWT at time, $i$. The MAE computes the average magnitude of the errors in a set of predicted values to obtain the average absolute difference between

the predicted, $P_i$ and the observed, $O_i$. In contrast to MAE, the MSE squares the error terms, resulting in the squared average difference between the predicted and observed values. The resultant MSE is not in the same units as the value of interest making it difficult to interpret. As the square root of the MSE, RMSE provides an error index in the unit of the data (Legates & McCabe, 1999). However, both the MSE and RMSE are more sensitive to outliers and less robust than MAE.

$$PBIAS = 100 \cdot \frac{\sum_{i=1}^{N}(P_i - O_i)}{\sum_{i=1}^{N} O_i} \qquad \text{eq. 4}$$

Another error index used in SWT modeling is called the percent bias (PBIAS) index. PBIAS computes the average tendency of model predictions to be greater or smaller than the observations/measurements (Gupta et al., 1999). A PBIAS value of 0 is best, and low-magnitude values (closer to 0), denote stronger model accuracy. Positive PBIAS values suggest

model underestimation, while negative PBIAS values suggest model overestimation (Moriasi et al., 2007).

**3.3 Model Performance Metrics: Dimensionless**

The Nash-Sutcliffe Efficiency (NSE, also called NSC, NS, or NASH), is a "goodness-of-fit" criterion that describes the predictive power of a model. Mathematically, the NSE is a normalized statistic that computes the relative magnitude of the variance of the residuals compared to the variance of the measured/observed data (Nash & Sutcliffe, 1970). Visually, the

NSE shows how well the observed versus simulated data fit on a 1:1 line.

$$NSE = 1 - \frac{\sum_{i=1}^{N}(P_i - O_i)^2}{\sum_{i=1}^{N}(O_i - \overline{O})^2} \qquad \text{eq. 5}$$

where $\overline{O}$ is the average value of $O_i$. To compute the Kling-Gupta Efficiency (KGE):

$$KGE = 1 - ED, \qquad \text{eq. 6}$$

$$ED = \sqrt{(r-1)^2 + \left(\frac{\sigma_P}{\sigma_O} - 1\right)^2 + \left(\frac{\mu_P}{\mu_O} - 1\right)^2}$$

where $r$ is the linear correlation coefficient between predictions and observations. The purpose of the KGE metric is to reach
a balance between optimal conditions of modeled and observed quantities being perfectly correlated (i.e., $r = 1$), with the same variance ($\sigma_p / \sigma_o = 1$) and minimizing model output bias ($\mu_p / \mu_o = 1$). The Kling-Gupta Efficiency (KGE) is based on a decomposition of NSE into separate components (correlation, variability bias, and mean bias), and tries to improve on NSE weaknesses (Knoben et al., 2019). Like NSE, KGE = 1 is a perfect fit between model simulations/predictions and observations/measurements. However, NSE and KGE values cannot be directly compared because each metric is influenced
by the coefficient of variation of the observed time series (Knoben et al., 2019).

The Willmott index of agreement, $d$, ranging from 0 to 1, is defined as a standardized measure of model prediction error where a value of 1 is a perfect agreement between measured and predicted values, and a value of 0 indicates no agreement:

$$d = 1 - \frac{\sum_{i=1}^{N}(P_i - O_i)^2}{\sum_{i=1}^{N}(|P_i - \overline{O}| + |O_i - \overline{O}|)^2} \qquad \text{eq.7}$$

The Akaike Information Criterion (AIC) is a selection method used to compare several models to find the best approximating model for the data set of interest (Akaike et al., 1973; Banks & Joyner, 2017; Portet, 2020). For details on the mathematical derivation and application of AIC, please see Banks & Joyner (2017), Portet (2020) and Piotrowski et al. (2021). The AIC equation version shown was developed for the least-squares approach (Anderson & Burnham, 2004):

$$AIC = N \cdot ln(MSE) + 2 \cdot K \qquad \text{eq.8}$$

where $N$ is the number of samples, $K$ is the number of model parameters + 1, and $MSE$ is obtained by the model, for the respective dataset, per stream (Piotrowski et al., 2021). The Bayesian Information Criterion (BIC) was developed for studies
where model errors are assumed to follow a Gaussian distribution (Faraway & Chatfield, 1998; Piotrowski et al., 2021). For other versions of BIC, please see Faraway & Chatfield, 1998.

$$BIC = N \cdot ln(MSE) + Kln(N) \qquad \text{eq.9}$$

Unlike other performance metrics, the AIC and BIC are unique in their ability to penalize the number of parameters used by a model, thus favouring more parsimonious models. For both the AIC and BIC, lower values of criterion point to a better model (Piotrowski et al., 2021).

**3.4 Performance Metrics for Most-Cited ML Statistics**

Reviewing ML studies focused on SWT modeling (Table S1, S2), the most-cited performance metrics were: RMSE (45
citations), NSE (25), MAE (18) and $R^2$ (17). Having reviewed the literature and in agreement with previous published
recommendations (Moriasi et al., 2007), we recommend that a combination of standard regression (i.e., $r$, $r^2$, $R^2$),
dimensionless (i.e., NSE), and error index statistics (i.e., RMSE, MAE, PBIAS) be used for model evaluation and reported
together in future publications. As part of our efforts to propose guidelines for easier comparison of ML performance across
SWT studies, we identified the range in reported values for these four most cited metrics and show the spread of values in
the training/calibration, testing and validation phases in box plot form.

We begin with the standard regression and dimensionless statistics, $R^2$ and NSE, both of which have an optimal value of
1. Figure 2 shows the median $R^2$ per ML model per model phase for the cited publications. For example, Foreman et al.
(2001) used an ANN to model SWT in the Fraser Watershed in British Columbia, Canada. Their model estimated 1995-1998
tributary and headwater temperatures and reported a median $R^2$ (fig. 2) of 0.93 for the training/calibration phase. Over the
review period, $R^2$ range (2001-2024) was 0.65-1.00. We note that for process-based modeling, acceptable $R^2$ values start
around $R^2 \sim 0.50$ (Moriasi et al., 2007). In stark contrast, ML models published between 2000-2024 exhibited significantly
higher $R^2$ values with a median of $R^2 \sim 0.93$, across 17 studies (fig. 2).

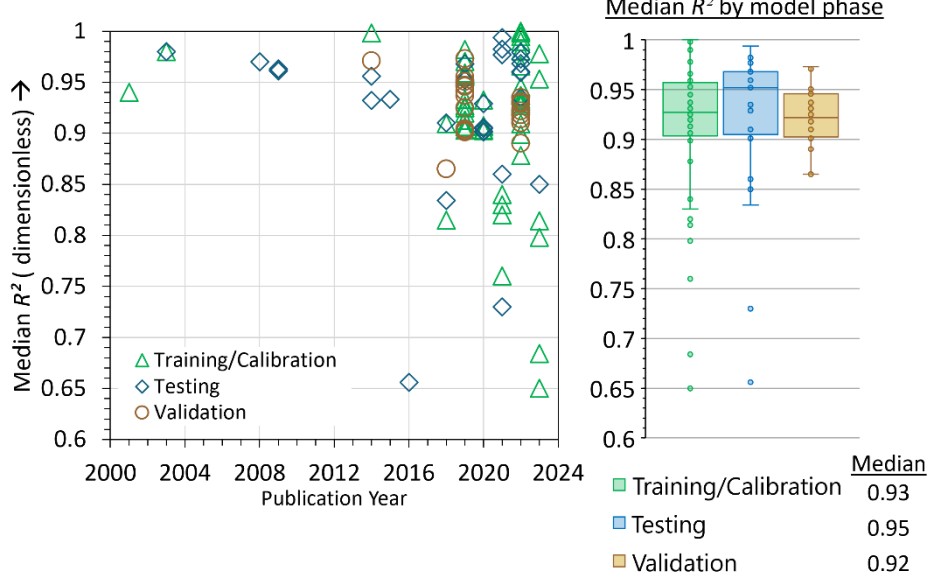

**Figure 2. Median $R^2$ (dimensionless) values from published literature for training/calibration, testing, and validation phases of
model evaluation.**

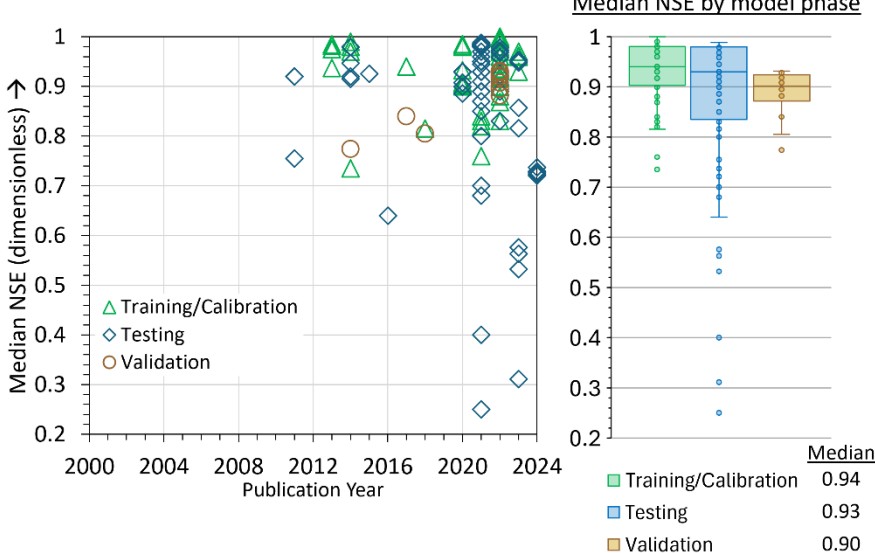

**Figure 3. Median NSE (dimensionless) values from published literature for training/calibration, testing, and validation phases of model evaluation.**

Unlike the $R^2$ metric, NSE was not used as a metric in ML studies of SWT between 2000 and 2010 (fig. 3). The first ML study to use NSE was St. Hilaire et al., (2011) to analyse SWT in Catamaran Brook, a small catchment in New Brunswick, Canada. Figure 3 shows that the NSE range reported by studies using ML for SWT was between 0.25-1.00 over the reviewed period (2000-2024). Like R2, NSE published values are high compared to traditional models (Moriasi et al., 2007, 2015), with a median NSE of 0.93 across 25 studies (fig. 3). Overall, these complimentary metrics should always be reported

together as they provide a broader evaluation of model performance, i.e. NSE measures a model's predictive skill and error variance, while R2 assesses how well the model explains the variability of the data.

Figure 4 shows the median RMSE (°C) and fig. 5 shows the median MAE (°C) per ML model per model phase for each publication. RMSE (°C) and MAE (°C) are popular error indices used in model evaluation because the metrics show error in the units of the data of interest (i.e., °C), which helps analysis of the results. RMSE and MAE values equal to 0 are a perfect

fit. Over the review period, median RMSE (fig. 4) ranged from 0.0002-3.50 °C. The median RMSE was 1.35 °C across 45 studies (fig. 4). Figure 5 shows that between 2000-2012, MAE was not used as a metric in ML studies of SWT. The first ML study to use MAE for SWT modeling was Grbic et al. (2013), where the Gaussian process regression (GPR) ML approach was compared with field observations of SWT from the Drava River in Croatia, to assess the feasibility of model development in SWT prediction. In contrast to RMSE, the MAE range (fig. 5) was 0.14-2.19 °C. The median MAE overall

was 1.09 °C respectively across 18 studies (fig. 5).

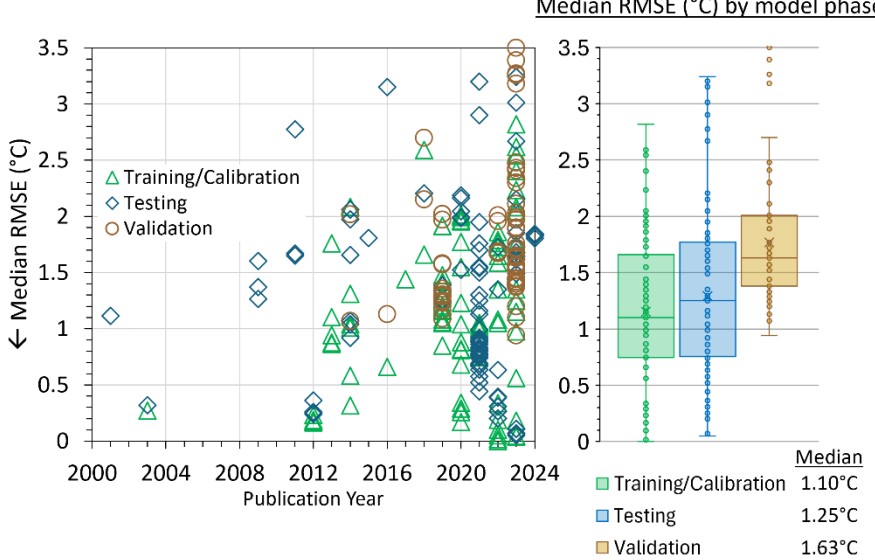

**Figure 4. Median RMSE (°C) values from published literature for training/calibration, testing, and validation phases of model evaluation.**

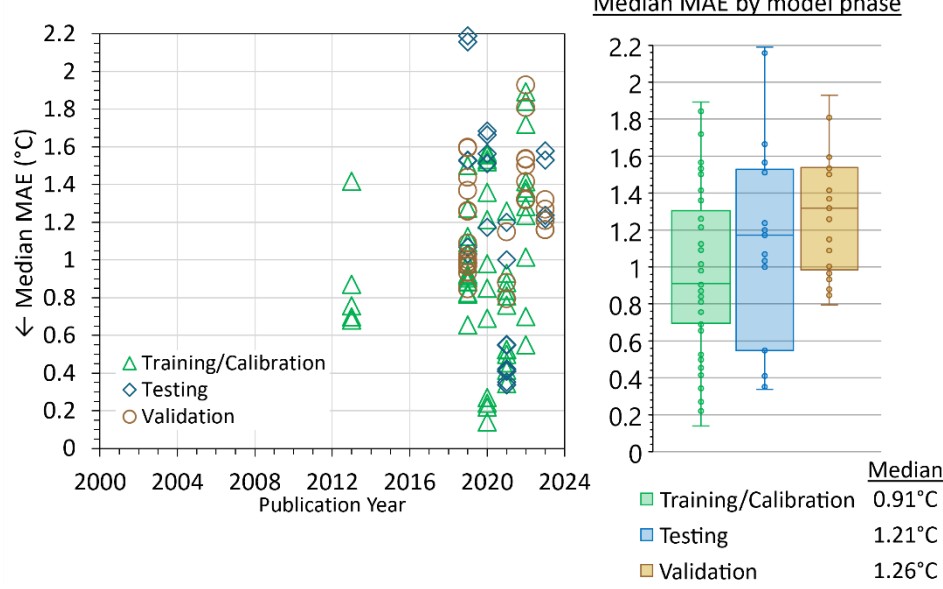

**Figure 5. Median MAE (°C) values from published literature for training/calibration, testing, and validation phases of model evaluation.**

**Table 1. Average, median, maximum, and minimum RMSE (°C) for studies grouped by local/watershed and regional/CONUS spatial scales.**

|  | Local/Watershed (< 100 km² area) | Regional/CONUS (> 100 km² area) |
|---|---|---|
| Number of data points | 900 | 1369 |
| Average | 1.52 | 1.55 |
| Median | 1.38 | 1.42 |
| Maximum | 5.170 | 4.387 |
| Minimum | 0.038 | 0.0002 |

## 3.5 Spatial Scale

We examined the data for the possible influence of spatial scale on the most-cited performance metric, RMSE, by grouping publications into two spatial categories: local, which included studies that focused on point to plot, specific sites, and small watersheds less than ~100 km² in area (about the size of a HUC-08), and regional, which included everything over ~100 km² in area. For this analysis, all RMSE values reported by publications were compiled into a table (not shown) and classified as belonging to either the local/watershed or regional/CONUS scale. A comparison of the data found that the average RMSE was similar for the local (~ 1.52 °C) and regional (~ 1.55 °C) categories. The median local RMSE was slightly better than the regional RMSE (~ 0.04 °C), but arguably within a standard of error. The local/watershed category had a higher maximum and minimum RMSE than those reported for the regional category. Overall, neither category appeared significantly better or worse than the other.

**Table 2. Suggested ratings for performance metrics (median) using metrics published by ML studies examining SWT.**

|  | $R^2$ | | | NSE | | |
|---|---|---|---|---|---|---|
| Rating | Training | Testing | Validation | Training | Testing | Validation |
| Very Good (>) | 0.99 | 0.99 | 0.96 | 0.99 | 0.98 | 0.93 |
| Good (range) | 0.89 - 0.99 | 0.92 - 0.99 | 0.94 - 0.96 | 0.92 - 0.99 | 0.84 - 0.98 | 0.88 - 0.93 |
| Satisfactory (range) | 0.79 - 0.92 | 0.86 - 0.92 | 0.91 - 0.94 | 0.85 - 0.92 | 0.70 - 0.84 | 0.83 - 0.88 |
| Unsatisfactory (<) | 0.79 | 0.86 | 0.91 | 0.85 | 0.70 | 0.83 |
|  | **RMSE (°C)** | | | **MAE (°C)** | | |
| Rating | Training | Testing | Validation | Training | Testing | Validation |
| Very Good (>) | 0.25 | 0.26 | 1.15 | 0.33 | 0.42 | 0.86 |
| Good (range) | 1.34 - 0.25 | 1.51 - 0.26 | 1.80 - 1.15 | 1.18 - 0.33 | 1.12 - 0.42 | 1.32 - 0.86 |
| Satisfactory (range) | 2.43 - 1.34 | 2.77 - 1.51 | 2.45 - 1.80 | 1.70 - 1.01 | 1.97 - 1.19 | 1.79 - 1.32 |
| Unsatisfactory (<) | 2.43 | 2.77 | 2.45 | 1.70 | 1.97 | 1.79 |

**3.6 Temporal Scale**

Across studies, there was large variability in the focus of temporal scales and use. For example, some studies used data collected at 5-15 minute intervals to simulate SWT at daily or weekly intervals for an abbreviated period (Risley et al., 2003; Hong & Bhamidimarri, 2012). Other studies used data collected at hourly, daily, weekly, or monthly intervals (Foreman et al., 2001; Sivri et al., 2007; Temizyurek & Dadaser-Celik, 2018) for periods of record spanning weeks (Lu & Ma, 2020; Abdi et al., 2021) to several decades (Cole et al., 2014; Weierbach et al., 2022; Heddam et al., 2022; Topp et al., 2023; Rehana & Rajesh, 2023), to simulate SWT. Concurrently, output for studies was then provided at resolutions ranging from hourly to monthly periods for the past, present or future. Given the use of study-specific temporal outputs and the limited amount of reported peer-reviewed model performance data at the temporal scales used by researchers, it was difficult to conduct statistical comparisons for temporal scales, so they are not further discussed in this review. We strongly suggest to researchers that metrics be made available at the temporal scale of interest (and not just for the overall model) in appendices or supplementary information to encourage more comparison across studies.

**4 Discussion**

**4.1 Model Evaluation Ratings**

From our review of RMSE, $R^2$, NSE, and MAE, we compiled ratings for ML performance metrics that should be used to for cross-comparison across SWT studies. From table S2, we note that there was not a consistent way of reporting training/validation/testing percentages, for example, some studies only reported performance metrics for one modeling phase (i.e., training), while others used "testing" and "validation" interchangeably, which could affect interpretation of model performance (Laanaya et al., 2017; Voza & Vuković, 2018; Hani et al., 2023). Additionally, others stated information not by percentages but by years (i.e., training 2 years, testing 1 year, validation 1 year), which can make comparisons challenging. Despite all the different ways that researchers chose to compile performance metrics, most models had strong metrics, as shown by our calculated ratings for performance metrics shown on table 2.

We posit that the definitions of satisfactory, good, and very good be updated to reflect the inherent capability of a ML algorithm to fit the input data more successfully than other model types, such as statistical models and process-based models. For example, $R^2$ values from ML-SWT studies that may appear to be very good, such as $R^2 \sim 0.91$, should be considered satisfactory given the context of the performance metrics published in ML-SWT literature. In table 2, the very good and unsatisfactory ranges were calculated from the box plots by identifying the two-thirds distance from the upper (or lower) quartile to the respective extreme whisker. This calculation identifies the ~8% of the data that is relatively closest to the minimum or maximum values of the box plots, indicating a very good or unsatisfactory value. For table 2, the separation between the satisfactory range and the good range was denoted as the halfway value between very good and unsatisfactory.

The purpose of these guidelines is to serve as a reference for SWT studies looking to understand and consider ML performance relative to other SWT-ML studies.

## 4.2 ML Data Requirements vs. Data Availability

While, in recent years, access to hydrologic data has improved (Miller et al., 2016; CUAHSI, 2024), data remains scarce for many hydrologic applications including SWT research, particularly because continual project management and funding to place and maintain stream temperature sensors, can be expensive and/or time-consuming to undertake. As a result, in the 21st century, the scarcity of data remains a large impediment for the application of machine learning in SWT modeling. What is more, the question of data quantity (how much data do you have?) versus quality (how much diverse data is needed?) continues to hinder ML use in hydrologic applications. Xu and Liang (2021) make the excellent point that one year of streamflow data (can swap for stream temperature) at 15-minute intervals equals about ~35,000 points, which may seem extensive, but is unlikely to be enough to properly train a ML model due to autocorrelation and limited exposure to diverse types of data that are naturally encountered with a longer time-series (Xu and Liang, 2021). For example, machine learning models may only predict flood volumes they have previously seen (Kratzert et al., 2019). While data requirements for ML remain high, there are some strategies that researchers have used to alleviate this impact.

One strategy that hydrologists in other fields have used to tackle this problem is data augmentation, which can be applied spatially or temporally to create new training examples that the ML model can learn from. Spatial augmentation can be done by means of interpolation methods, i.e., kriging or distance weighting to create new data points or by generating synthetic data based on expected physical patterns to fill gaps in data coverage (Baydaroğlu & Demir, 2024). Temporal data augmentation can be done by shifting, scaling or adding noise to existing time series to create new training examples (Skoulikaris et al., 2022). Alternatively, and not a new idea, is to use the statistical technique known as seasonal decomposition, which breaks down a time series into its main components, i.e., the trend, seasonal patterns and residual components (Apaydin et al., 2021; He et al., 2022). These can then be recombined to generate new data and train the model for improved accuracy (Apaydin et al., 2021). In addition to data augmentations, data requirements can be alleviated by considering the help of unsupervised transfer learning, i.e., use pre-trained models on similar tasks to reduce amount of data needed for training, or semi-supervised learning, such as few shot learning, i.e., combine a small percent of labeled data with larger percent of unlabeled data to improve model performance (Yang et al., 2023). By implementing these strategies, researchers in other hydrologic fields have shown that models can be improved with less data, strategies that are likely transferable to SWT research.

## 4.3 ML Use for Knowledge Discovery

It has been suggested that the increasingly prominent use of ML for hydrological predictions points to a paradigm shift, one where the adoption of ML in most if not all future physical hydrologic modeling appears certain (Xu & Liang, 2021; Varadharajan et al., 2022). As physical scientists try to stay afloat in a sea of ML algorithm options and processes, there is a

critical need to examine how "newer" tools such as ML are improving our understanding of the natural world. Our review finds that ML studies examining SWT have been conducted from a computational perspective, one with a focus on comparing techniques and performance as opposed to explaining the nature of SWT dynamics or influencing processes.

While it is understandable that not every ML-SWT paper aims to explain physical processes, the SWT community should agree on a baseline of tests that all ML-SWT models undergo to assess model robustness and transferability. Specifically, we urge use of TUURTs (Temporal, Unseen, Ungaged Region Tests) for future ML-SWT models as a helpful step towards better modeling practices, increased model transparency and robustness (see section 2.4.1, Fig.1). From a computational perspective, the use of ML in SWT modeling has led to improvements in pattern identification (i.e., release of

water from reservoirs; see Jiang et al., 2022) and examination of climate events (i.e., extreme droughts; see Qiu et al., 2020), with the aid of observations and remote sensing data. The use of ML for estimating hydrologic variables (i.e., precipitation, snow water equivalent, and evapotranspiration) and approximating hydrologic processes (i.e., runoff generation) has also become increasingly common due to the ML's ability to use many inputs without the bounds of preexisting relationships (Xu & Liang, 2021). In addition, hybridizations that couple ML models (i.e., WT-LSTM, K-nn with LSTM; Rehana and Rajesh,

2023) or couple ML with process-based models (SNTEMP-LSTM, Rahmani et al., 2023) show potential for outperforming extensively calibrated hydrologic models, especially where physical constraints can be introduced (Rahmani et al., 2023).

Recent studies (Rahmani et al., 2023; Wade et al., 2023), have tried to infer drivers of SWT regimes by accounting for some level of physics. Compounding the challenge of applying physical laws without negatively affecting the performance of a ML model, is the problem that the ML model itself is not immune to the difficulties met by statistical and process-based

models such as: data uncertainties, parameter uncertainties, and equifinality (Beven, 2020; Varadharajan et al., 2022). These uncertainties, coupled with the alarming trend of consistently high marks of the performance metrics discussed here, point to an imperative need to reevaluate how best to use ML in a manner that addresses knowledge gaps of physical systems instead of perfecting performance that is unlikely to be insightful for physical processes and trends. Our review of the literature and analysis of the performance data agrees with discussion by Beven (2020), who examined the future of hydrological sciences

with ML, and posed several important questions regarding better use of ML models for scientific inquiry.

## 4.4 Future Directions of SWT Modeling

The utility of ML in hydrologic modeling has advanced significantly, with interest seemingly growing exponentially (Nearing et al., 2021). With the novelty of ML, it is easy to over-value model performance and ignore the physics of the system, but with several decades of ML-experience, we advocate it is necessary to purposefully use ML to address

physically meaningful questions and not just create ML for the sake of creating. Given this, Varadharajan et al. (2022) laid out an excellent discussion on opportunities for advancement of ML in water quality modeling, see section 3 of publication of Varadharajan et al. (2022). (Varadharajan et al., 2022). Here we highlight some of the questions from Varadharajan et al. (2022) that can be considered in the context of what objectives the SWT community should be using in the ML era, namely: 1) How do we use physical knowledge (re: heat exchange equations, radiation influence) to improve models and process

understanding? Rahmani et al. (2023) coupled NNs with the physical knowledge from SNTEMP, a one-dimensional stream temperature model that calculates the transfer of energy to or from a stream segment by either heat flux equations or advection, but found that even with SNTEMP, their flexible NNs exhibited substantial variance in prediction and needed to be constrained by further multi-dimensional assessments (Rahmani et al., 2023). In short, if our use of physics in machine learning makes our models worse, we should understand why.

A second question that needs addressing is 2) How do we deal with predictive uncertainty in ML used for SWT modeling? According to Moriasi et al. (2007), uncertainty analysis is the process of quantifying the level of confidence in any given model output based on five guidelines: 1) the quality and amount of observations (data), 2) the lack of observations due to poor or limited field monitoring, 3) the lack of knowledge of physical processes or operational procedures (instrumentation), 4) the approximation of our mathematical equations, and 5) the robustness of model sensitivity analysis and calibration. For example, in rainfall-runoff modeling, researchers have proposed benchmarking to examine uncertainty predictions of ML rainfall-runoff modeling (Klotz et al., 2022). For stream temperature modeling, researchers have attempted to address the role of uncertainty in deep learning model (RGCN, LSTM) predictions using the Monte Carlo Dropout (Zwart, Oliver, et al., 2023) and a unimodal mixture density network approach (Zwart, Diaz, et al., 2023).

Other questions that SWT-ML studies should consider are: 3) How do we make ML models generalize better, specifically with regards to ungaged basins? And 4) How can ML models be improved to predict extremes? As ML models advance to use satellite data, include more sensor networks and/or couple with climate models, there is a logical next step toward creating generalizable models that can account for extremes. The challenge of prediction in ungaged basins in SWT modeling has been explored for at least a decade by process-based (Dugdale et al., 2017) and statistically based (Gallice et al., 2015, Isaak et al., 2017; Wanders et al., 2019; Siegel et al., 2023) models. Unfortunately, process-based models continue to be limited by data requirements and memory or processing/programming impediments (Dugdale et al., 2017; Ouellet et al., 2020), while statistically based models struggle to account for changing physical conditions (Benyahya et al., 2007; Arismendi et al., 2014; Lee et al., 2020). Physics-derived statistically based models have been applied in ungaged regions (Gallice et al., 2015) but models tend to be region-specific and not generalizable. We posit that a future direction of ML models is to expand on their ability to learn, identify and mimic the complexity needed to improve SWT predictions for ungaged basins. To date, researchers have used ML to model SWT for partially ungaged (i.e., discharge used as input) regions across the CONUS (Rahmani et al., 2020, 2021), though limitations persist in hydrologically complex and critical regions in the west (CA) and southeast (FL). Recently, a satellite remote sensing paper used RF to model monthly stream temperature across the CONUS and tested for temporal (walk-forward validation), unseen and 'true' ungaged regions (Philippus, Sytsma, et al., 2024). Given community-wide modeling interest expanding from SWT prediction to forecasting (Zhu and Piotrowski, 2020; Jiang et al., 2022; Zwart, Diaz, et al., 2023), ML-use could prove essential in capturing unknown, complex SWT patterns in space and time (Philippus, Corona, et al., 2024) and with shifting baselines. With regards to ML models such as LSTMs predicting extremes, a limitation that must be addressed is that they generally only make predictions within the bounds of their training data (Kratzert et al., 2019) though researchers are looking to improve on

this by using ML-hybridizations (Rozos et al., 2023). Overall, there is promising work in the community towards creating ML models for SWT that generalize better and/or are more robust for predictions of extremes.

Finally, 5) How can we build ML models such that they are seen as trustworthy and interpretable by the hydrologic community? To answer this question, we must address a technical barrier (black-box issues, data limitations, model uncertainty) and a social barrier (i.e., educated skepticism of ML due to novelty, little understanding of computer science basics and/or coding experience). If we are to incorporate ML into decision-making processes, it makes sense that ML must be transparent and understandable to more than just computer or data scientists (Varadharajan et al., 2022). For example, Topp et al. (2023) recently used explainable AI to elucidate how ML architectures affected the SWT model's spatial and temporal dependencies, and how that in turn affected the model's accuracy. Addressing this technical barrier can also be done by improving access to data, which has seen remarkable progress thanks to web repositories such as NSF-funded CUAHSI's Hydro share (CUAHSI, 2024) and GitHub (GitHub, 2024). In the United States, data access to state and locally based data remains limited and should be addressed. In terms of the social barrier, education about ML and ML-use is key.

Societal interest in ML has thankfully also led to a plethora of educational resources and ML walk-through videos and tutorials in Tensorflow (Abadi et al., 2016), PyTorch (Paszke et al., 2019), and Google Colab (Bisong, 2019). With the speed at which ML-use is evolving, short communication pieces (Lapuschkin et al., 2019) and opinion pieces (Kratzert et al., 2024) with clear examples about an ML-issue and practical solutions will also help make ML challenges more transparent and therefore accessible to the hydrologic community-at-large.

## 5 Conclusion

While initial examination of SWT began with statistical and process-based modeling many decades ago, there is now a strong interest within the hydrology community to use ML across the board to further our understanding of hydrologic causes and effects. Indeed, extensive progress has been made in using ML for SWT modeling solely in the last quarter century (2000-2024). As discussed in this review, applications of ML in SWT modeling have ranged from the local to the continental scale, as well as from the short-term period of hours to the longer-term period of decades.

In this review, we examined published literature that used ML for SWT modeling and provided a range of background information on the ML models used in these studies. Additionally, we compiled reported ML performance metrics and compare those most cited -- RMSE, $R^2$, NSE and MSE. We find that ML performance metrics surpass all our pre-conceived notions of what makes a very good vs. satisfactory model. We argue that as a scientific community, we need to redefine model success in the face of ML's consistently robust performance, or at the very least, hold ML to additional standards when comparing ML to physically based and statistically based models. To aid in re-defining standards, we introduce updated designations (for ML studies only) of very good, good, satisfactory, etc. performance metrics as derived from the literature. In addition to levelling the playing field when comparing ML results to process-based and statistically based

models, we assert that raising the performance bar could also strengthen user confidence in ML-models to the point that their consideration in decision-relevant predictions becomes more widely trusted and accepted.

Finally, our review finds that the increased accessibility to ML and its use in SWT modeling has yet to lead to better physical understanding of SWT causes and effects. Over the past 25 years, the focus on desired accuracy and performance metrics has overpowered much-needed trade-offs that earlier models of the 20$^{th}$ century considered, such as process

complexity (scale, heterogeneity, generalizability), knowledge discovery, timeliness, and basic public understanding. Given our knowledge that most ML models consistently perform at a higher level, we believe it is time to take a step back and purposefully consider more thoughtful creation and purposefulness of ML models for the goal of decision-relevant predictions that include risk-mitigation, water resource planning and process understanding of stream water temperature influencers and effects.

**6 Appendix A. Traditional artificial neural networks, detail and descriptions**

ANNs are composed of networks of interconnected neurons, also called nodes or units. The network architecture of a commonly used ANN, the *feed-forward NN (FFNN)*, can be described as a three- (or more) layered network of connected neurons, organized from left-to-right, where the input layer is the first layer, the centre layer (could be one or more) is "hidden", and the last layer is the output layer (Risley et al., 2003). *Multi-layer perceptron NNs (MLPNN)* fall under the

945 umbrella of FFNNs. In the FFNN architecture, the first (left-most) layer creates input signals from a dataset. In the hidden layer, the neurons process the input signals using an activation function (i.e., step, sigmoid-shaped, hyperbolic tangent, etc.), to calculate a hidden-layer output from the input, the hidden-layer weight, and the hidden-layer bias (Hinton, 1992). The hidden-layer weight is defined as the strength of the influence of neurons on each other and is modifiable (Hinton, 1992). For example, a connection between neurons A and B may be stronger (weight ~0.5) than a connection between neurons B and C

(weight ~0.1). This weight can be adjusted, or "fine-tuned" to minimize errors. Depending on the output of the activation function, the output signals may be transmitted to other neurons in the network, eventually supplying output from the hidden layer to the final layer, which computes the final output using a summation function (Hinton, 1992).

The backpropagation (BP) learning algorithm (Hinton, 1992) is one of the more popular techniques that iteratively

adjusts model weights and bias terms in a neural network. First, the FFNN is trained on a labelled/categorized dataset, called the "training" dataset. The BP algorithm then iteratively adjusts weights in the NN based on the calculated error between the predicted output and the actual output, allowing the NN to find underlying patterns or possible relationships in the data (Hinton, 1992). However, use of the BP learning algorithm for FFNNs can be time-consuming in terms of training and calibration (Huang et al., 2006). Huang et al. (2006) proposed an alternative learning algorithm called *extreme learning*

*machine (ELM)* for shallow-layer BP FFNNs (also abbreviated to BPNNs). The ELM algorithm optimizes training by

randomly choosing hidden nodes and analytically finding output weights (Huang et al., 2006). In a comparison study, ELM generally outperformed the BP algorithm in terms of learning and performance (Huang et al., 2006).

Another kind of ANN with a similar three-layer structure is the *radial basis function NN (RBFNN)*. However, the RBFNN distinction is that only one hidden layer is used and that the width of connections and centres (distance between inputs and weights) must be calculated prior to adjusting weights (Musavi et al., 1992; Buhmann, 2000). We refer the reader to Musavi et al. (1992) and Buhmann (2000) for more detail on RBFNN. The *Cascade Correlation Neural Network (CCNN)*, introduced by Fahlman and Lebiere (1989), proved to be much faster than back-propagation (Fahlman & Lebiere, 1989). The CCNN was created with a cascade architecture, where hidden neurons are added to the network one at a time and remain unchanged, i.e., the input weights are frozen, allowing the neuron to become a feature-detector in the network, capable of either producing outputs or creating other, complex feature detectors (Fahman and Lebiere, 1989). For more detail on CCNN, we refer the reader to Fahman and Lebiere, 1989.

*General Regression NN*, is a Bayesian type of FFNN based on kernel regression networks (Specht, 1991). Unlike MLPNN, GRNN does not need an iterative training procedure like backpropagation. One of the advantages of GRNN with increasingly larger datasets is that it is consistent in forcing the estimation error to approach zero with only minor restrictions on the function (Specht, 1991). GRNN also differs from RBFNN in the method used to decide the weights of the hidden layer nodes. GRNN does not train the weights as RBFNN does, instead, GRNN provides the target value (to the node weight) by considering the input training data set and the related output (Specht, 1991). The *Product-Unit NN (PUNN)*, uses product units (in contrast to the summation units used by MLPNN), to compute the product of its inputs, each raised to a variable power (Janson & Frenzel, 1993). While less used in SWT modeling, PUNNs have garnered interest due to their capacity for implementing higher order functions (Martínez-Estudillo et al., 2006), and advantage of requiring less parameters for optimization, when considering the same number of input nodes, hidden nodes, and output nodes (Piotrowski et al., 2015). For more on PUNN, we refer the reader to Janson and Frenzel (1993) and Martínez-Estudillo et al. (2006). A lesser known but used ANN is the *Group Method of Data Handling (GMDH)*, created by Russian scientist Ivakhnenko in the late 1960s for the purpose of using inductive learning methods for modeling complex, non-linear systems without the bias of the user (Ivakhnenko, 1970). Although not initially described as an ANN, GMDH is a polynomial NN. GMDH initiates only with input neurons, then during the training processes, neurons are "self-organized" to optimize the network with the help of "control data" to stop the training process when overfitting occurs (Ivakhnenko & Ivakhnenko, 1995; Graf & Aghelpour, 2021). For more information on GMDH, we refer the reader to Ivakhnenko (1970) and Ivakhnenko & Ivakhnenko (1995).

*Adaptive-network-based fuzzy inference systems (ANFIS)* are a type of NN using fuzzy inference, initially proposed by Jang (1993). Fuzzy inference systems first interpret values in the input vector, then (following a set of rules), the system assigns values to the output vector (Kalogirou, 2023). ANFIS uses a combination of fuzzy inference, and adaptive network

learning (a superset of all FFNNs) to make and improve upon its estimations (Jang, 1993). In SWT modeling, ANFIS has been included in comparisons with other ANNs for model performance evaluation (Piotrowski et al., 2015; Zhu, Heddam, Nyarko, et al., 2019; Zhu, Hadzima-Nyarko, Gao, Wang, et al., 2019; Graf & Aghelpour, 2021). A different type of fuzzy ANN is the *dynamic neuro-fuzzy local modeling system (DNFLMS)*, which contrasts with ANFIS by its use of the one-pass clustering algorithm and sequential learning algorithm (Hong & Bhamidimarri, 2012). A comparison of ANFIS and

DNFLMS showed that the latter requires less training in terms of fuzzy rules needed and less epochs, which can result in over 18.5 hours saved in computing time (Hong & Bhamidimarri, 2012).

## 7 Code Availability

Computer code was not used to conduct this review.

## 8 Data Availability

All data was obtained from the cited publications. Data used for figures 1 to 4 and tables 1 and 2 are available online at CUAHSI HydroShare: http://www.hydroshare.org/resource/68edf1673096480bacc6bd104215e9dc

## 9 Supplement link

Copernicus will provide.

## 10 Author contribution

C.R. Corona contributions included: conceptualization, data curation, formal analysis, investigation, methodology, validation, visualization, writing - original draft preparation and writing - review & editing. T.S. Hogue contributions included: conceptualization, funding acquisition, methodology, project administration, resources, supervision, writing - review & editing.

## 11 Competing interests

The authors declare that they have no conflict of interest.

## 12 Disclaimer

This manuscript and related items of information have not been formally disseminated by NOAA, and do not represent any agency determination, view, or policy.

## 13 Acknowledgements

This project was supported by the NOAA Cooperative Institute for Research to Operations in Hydrology. Funding was awarded to Cooperative Institute for Research to Operations in Hydrology (CIROH) through the NOAA Cooperative Agreement with The University of Alabama (NA22NWS4320003).

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
