# Peer review of "Machine Learning in Stream/River Water Temperature Modeling: a review and metrics for evaluation"

_Hydrology and Earth System Sciences, 2024_

## Referee Comment (RC1)

The manuscript on "*Machine Learning in Stream/River Water Temperature Modeling, a review and metrics for evaluation*" focuses on providing a comprehensive review of Machine Learning studies, including traditional and recent methods in ML and AI, on stream temperature modeling and prediction. Overall, the manuscript is well-written and covers most of the relevant papers, but there are a few strategic points I would like to share with the authors:

- Figures 1 & 2 & 3 & table 2: The manuscript provides a table for multiple metrics such as R2, NSE, RMSE, and MAE, and suggested a rate of numbers to rate the ML methods' performances. This table is based on the metrics that have been achieved by the studies in the previous years which are reflected in figures 1 & 2 & 3. However, those studies vary in terms of case studies, number of basins included in the study, running regional or local models. We know that ML models are prone to overfitting, especially for stream temperature that follows a relatively sinusoidal curve through a year, which means it is more predictable for complex models such as LSTM. However, it means the models are prone to easily overfit. Therefore, I suggest the authors encourage the stream temperature researchers to go towards making more generalizable models and less overfitted. For example, instead of suggesting performance metrics, the authors can provide a few steps to make sure the models are not overfitted or underfitted. For instance, always considering a spatial test on ungauged sites (basins). We know that spatial tests are more difficult tasks rather than temporal tests. Therefore, it is acceptable to get lower performance on ungauged basins, however, the metrics should not be very different from temporal tests. A more challenging experiment is to test the trained model on regions that have not been seen by the model. In theory, if a model has been able to capture true relations between the driving factors on stream temperature, it should achieve a relatively decent performance on basins

with different hydrologic, geologic, and climatic characteristics from the trained basins. As a researcher on stream water temperature, I would rather to have a model that passes all these three tests (temporal, ungauged, unseen regions) with relatively close metrics, rather than having a model that gives very high performance in temporal tests and low performance in the other two tests.

- Evaluation of Data Requirements: The manuscript does not extensively discuss the challenges that ML stream temperature modelers are facing with. Different ML models have varying data requirements, but the review does not thoroughly discuss the data needs for each type of model. For example, machine learning models are dependent on data. If we compare the availability of streamflow observation data availability versus the stream water temperature observation data, we realize there is a massive gap here, which impacts the studies and reduces the SWT model performances. I suggest, while the authors encouraging the researchers and water institutes to collect more data, they add their comments on this issue and discuss how researchers can reduce the impact of this problem in their models.

- Future Directions Could Be Expanded: Although the paper concludes with a general discussion of future challenges, it does not offer specific, actionable directions for future research. Highlighting key areas where ML can advance, such as the use of satellite data, sensor networks, or the fusion of climate models with ML, would provide more meaningful insights. In this concept, we can learn from hydrologic community and capitalize on their experience and what they learned. The ML hydrologic community is moving toward making global models, incorporating mechanistic models into their ML framework and learning the governing factors, flow prediction with predicted inputs (predicted

meteorological inputs) and last but not least, providing a seamless simulation in streams in CONUS/global scale.  Therefore, I would ask the authors to add their comments on where the future direction of SWT community should be and  how SWT community can achieve the future objectives and what the barriers are.

- The manuscript walked through many ML and AI models. An important factor of the ML and AI models are the inputs. I assume you faced a variety of inputs that have been used in the models. That would be informative to the readers, if the authors add their observations that what kind of inputs that have been missed to be used, either because it is not available yet or it is even missed. For instance, whether there is any geophysical attribute, climatic attributes, or any forcings that is worth to be extracted and used in ML models.

- Lack of Clear Structure in the Evaluation: Although the paper aims to summarize the performance evaluation metrics for ML models in SWT prediction, the organization of these sections feels somewhat scattered. A more systematic approach could improve clarity, such as separating the analysis based on time scales (e.g., hourly, daily, monthly) or spatial scales (local, regional, continental). This would make it easier for readers to find the relevant insights based on their application. For instance, a stream temperature model in monthly scale is different from a daily or hourly scale models on many aspects. As an example, the complexity of a daily model is different from a monthly temperature models. A monthly model may not need all inputs of a daily model to capture the monthly changes. The authors can add their overall opinion of what types of models are better fitted to which time scale. In ML models, it is important to know the scope of the model, whether it is a local model that needs to be calibrated site by site, or it is a model that  is designed to work

for multiple sites (a regional model). I believe that would be informative to consider the modeling approach when methods are compared.

- I believe the authors need to decide first who are the readers of the papers. Whether the paper serves to new-commers to ML and AI methodologies in stream temperature community or it serves to researchers that are already familiar with basics of ML and AI methods. While the paper provides an extensive review of machine learning (ML) applications in stream water temperature (SWT) modeling, it focuses heavily on listing the types of ML models used rather than deeply analyzing their applications, strengths, weaknesses, and performance differences. A more critical analysis of the pros and cons of each model type could provide greater value to researchers choosing the appropriate model for their specific needs. To provide a few examples, I refer you to lines 136 – 143 & lines 146 – 159 & lines 263 - 292. The first half of the paragraph that is written in lines 136 – 143 explains the fundamentals of the method, which may not be necessary to be long, and the rest is an example of the method usage. However, this paragraph could have been enriched by statements like the advantages and disadvantages of this method compared to other existing ML methods or even to a linear regression method, or a 1D mechanistic method (although they are not ML methods, but the comparison is beneficial to the readers). The authors also can add their statement of under what conditions they think the method is beneficial. Lines 146 – 153 explains PCA and ck-means clustering on data reduction application, however, it is not clear here under what conditions we can use them. Additionally, that would be nice for readers if the authors add feature importance to their comparison as it has been used more frequently in streamflow and soil moisture prediction studies. Lines 263 – 292 are organized in three paragraphs while providing general

knowledge about ANNs with relatively less direct relations to water temperature application.

- Line 13: There is a typo that changes the meaning of the sentence. It should be "… with in situ …" or "… with in-situ …".

- Line 132; There is a typo here too. It should be "long short-term memory". Although I am trying to catch them, there is a chance that I miss some of them. I recommend the authors to carefully re-read the manuscript or ask help from a fresh pair of eyes to find these types of typos.

- Lines 208 – 210: to make the sentence more accurate, it needs to be stated whether these are local models or one model for multiple sites. Additionally, I believe by "NNs" here, the authors mean feedforward neural network, which are totally different from recurrent neural networks.

- Line 541 : "at" is missed. It is .. All journals examined used at least …"

---

## Author Comment (AC1)

**Referee #2 Comments**

This is a meaningful manuscript that provides a thorough review of ML approaches for SWT modeling and their evaluation metrics. I believe that the current scientific community has indeed developed a broad understanding of the integration of ML into stream temperature modeling. Hence, while the manuscript presents a comprehensive overview, incorporating more in-depth insights could enhance its appeal to readers and significantly increase its contribution to the field. The review covers a wealth of content, including recent articles and other reviews, but the sections are somewhat loosely structured, with key points relatively briefly mentioned.

**AUTHOR RESPONSE:** We thank the referee for their time and feedback, we believe the manuscript is stronger as a result. We address specific referee comments below. For reference, we separated some referee comments into a, b, etc., to provide a more organized response.  Proposed new/edited text is in BLUE.

**1.** For instance, in the first section (Overview: SWT Model Types), the author provides a solid overview of statistical, physical, and ML models. However, a more detailed analysis of the comparative strengths and weaknesses of physical and ML models would strengthen the discussion. The models are presented in a nearly linear developmental order in this review, but it would be beneficial to mention some points, for example, [if] physical models perform well, why ML models are adopted[?].

**AUTHOR RESPONSE**: The referee makes a good point with regards to the question of "if physical models perform well, why are ML models being adopted?". We have expanded the section "Artificial Intelligence Models in SWT Modeling" to discuss this:

"In the last decade, computing advances in AI have started to offer several advantages for using machine learning (ML) in hydrology that are comparable to physically based models (Cole et al., 2014; Zhu et al., 2019; Rehana and Rajesh, 2023). In contrast to traditional physically based models, the code underlying ML models are generally open-source and publicly available allowing for near real-time accessible advances and user feedback, whereas the source code for some physically based models may be inaccessible to the public due to being privately managed (MIKE suite of models) or the model software may be publicly available but take years to publish updates (USGS MODFLOW, Simunek's HYDRUS). One advantage that has made ML increasingly appealing includes its ability to learn directly from the data (i.e., data driven), which can be useful when the underlying physics are not fully understood or are considered too complex to model accurately.

Additionally, ML models are more efficient in making predictions compared to the time-intensive solvers of physically based models. ML models can also handle the challenge of scalability, that is managing large datasets and seamlessly deploying across various computer platforms and applications (Rehana and Rajesh, 2023). Air2stream, a hybrid statistical-physically based SWT model (Toffolon and Piccolroaz, 2015; Piccolroaz et al., 2016), initially outperformed earlier ML models such as Gaussian Process Regression  (Zhu et al., 2019). Though in the last few years, Air2stream has had its performance matched and even exceeded by recent neural networks models (Feigl et al., 2021; Rehana and Rajesh, 2023).

Finally, with computer processing power improving and the emergent field of quantum computing, there is a strong belief amongst scientists, stakeholders and the public, that using ML and by extension AI, in science applications will drive innovation to the point where natural patterns and insights not currently apparent in physical modeling will be uncovered (Varadharajan et al., 2022). Thus, while physically based models are considered tried-and-true, thereby invaluable for their interpretability and grounding in established physics, ML models have the potential for growth – where they can be used to first complement and eventually lead as powerful tools for prediction, optimization, and understanding in increasingly complex and data-rich environments."

New citation:
Toffolon, M. and Piccolroaz, S., 2015. A hybrid model for river water temperature as a function of air
temperature and discharge. *Environmental Research Letters*, *10*(11), p.114011.

**2.** How to gain the trust of traditional model users in ML methods? (This question is inherently
challenging, as model users often have preferences based on their own familiarity with certain models and
may exhibit biases against alternative approaches. However, it may be worthy to acknowledge this in the
review.) This discussion could extend to the choice between different ML models as well, as conclusions
favoring one model over another often depend on the specific context of the study. Many conclusions are
applicable only under particular circumstances, so a generalization such as "a certain model is better
suited to a particular type of problem" is more appropriate.

**AUTHOR RESPONSE:** We agree and appreciate the referee's feedback. We address this comment in
our response to referee #1 for comment #3 (copied below) titled "Future Directions", where we discuss
how researchers can work to present their ML models as trustworthy. For this, we propose adding a new
'Discussion' subsection, titled 'Future Directions of SWT Modeling', with the following:

[revised manuscript text omitted]

**AUTHOR RESPONSE:** We agree. Referee #1 made a similar comment (ref #1, comment #1A) about
overfitting and having ML undergo more testing and we propose to address both comments by adding: 1)
a subsection under "section 2.4 SWT Predictions using ML" on overfitting/underfitting and 2) a diagram
showing initial steps to mitigate overfitting. The new text is below:
Subsection 2.4.X Overfitting and Underfitting
When a model is too complex, i.e., has too many features or too many parameters relative to the
number of observations, or is forced to overextend its capabilities, i.e., make predictions with
insufficient training data, the model runs the risk of overfitting (Srivastava et al., 2014). An
overfitting model fits the training data "too well", capturing noise and details that provide high
accuracy on a training dataset, only to perform poorly once the model encounters "unseen" data in
testing/validation (Xu and Liang, 2021). Scenarios where overfitting may be temporarily acceptable
are those where: 1) model development is at its preliminary stages, where the interest is in a "proof of
life" concept, 2) when the objective is to identify heavily-relied on features by the model, i.e., feature
importance, or 3) in highly-controlled modeling environments where the expected data will be
consistently similar to the training dataset. The latter is more likely in certain industrial applications
and unlikely in the changing nature of hydrology.
In contrast, underfitting occurs when a model is too simple to capture any patterns in the data,
which can also lead to terrible performance in training, testing and validation. Underfitting can occur
with inadequate model features, poor model complexity or when regularization techniques, (e.g., L1
or L2 regularization), are over-used, making the model too rigid and unable to respond to changes in
the data. Given the propensity of machine learning models to effectively learn the training data, underfitting is less of an issue in ML whereas overfitting can be widespread. In the following
diagram, we present an example workflow to transition away from overfitting and towards
generalizability. We further encourage modelers to actively transition towards making more
generalizable models, which are in theory, more capable of performing well across diverse scenarios
and datasets, which will become increasingly important with the persistence of climate extremes.

[Figure]

Figure XX. Diagram showing steps that can be taken in modeling process to mitigate overfitting.
With regards to generalization, we propose to address this comment and a similar one made by ref #1
(comm# 3) by adding a new Discussion subsection, titled 'Future Directions of SWT Modeling'. Below is
our response in that section (full section is copied at comment #2) about generalization:

Another question that SWT-ML studies should consider is 2) How do we make ML models
generalize better, specifically with regards to ungaged basins? And 3) How can ML models be
improved to predict extremes? As ML models advance to use satellite data, incorporate more sensor
networks and/or couple with climate models, there is a logical next step towards creating
generalizable models. In our review, only two papers (Rahmani et al., 2020, 2023) conducted a
CONUS-scale approach towards SWT-ML modeling, but omitted large parts of the southwest (CA)
and southeast (FL), two hydrologically important regions. Recently, a satellite remote sensing RF was
used to model monthly SWT across the CONUS and tested for temporal (walk-forward validation),
unseen and 'true' ungaged regions, with the model architecture potentially generalizable due to it not
being location-specific (Philippus et al., 2024). We have also learned that certain ML models such as
LSTMs, can only predict within the bounds of their training data (Kratzert et al., 2019), which is a
limitation for predicting extremes. Thus, we strongly urge the community to work towards ML
models that generalize better and/or are more robust towards predictions of extremes.

**3b.** Additionally, the review does not systematically address the critical issue of model input selection,
which is essential in ML modeling. Model inputs for SWT modeling may include hydrometeorological
and physical parameters (or other attributes used in different studies), they play a role in model
performance and should be discussed in this part.

**AUTHOR RESPONSE:** Thank you for pointing out this area in need of clarity. Referee #1, comment #4
had a similar question about model input, and we propose adding the paragraph below in response to
both. Additionally, we want to note that we included in Supplementary Materials, Table S1, which
contains some of the suggested data by the referee, such as: period considered, region examined, temporal
resolution of SWT, spatial scale of study, and hydrometeorological parameters used for modeling.
This section will likely precede the "Local" and "Regional" subsections of 2.4 and be titled Model Inputs
for ML-SWT:
Using air temperature (AT) to better understand SWT has been considered since at least the
1960s, when Ward (1963) and Edinger et al. (1968) discussed the influence of air temperature on
SWT. Since then, studies have used varying input variables (see Table S1), however, the model inputs
of AT and SWT continue to be the most used in ML-modeling studies. In particular, studies have
used AT from time periods outside of the known SWT record to improve model performance (Sahoo
et al., 2009; Piotrowski et al., 2015; Graf et al., 2019). In addition to AT and SWT, flow discharge has
been used to attempt to constrain SWT (Foreman et al., 2001; Tao et al., 2008; St-Hilaire et al., 2011;
Grbić et al., 2013; Piotrowski et al., 2015; Graf et al., 2019; Qiu et al., 2020). Traditionally-used
model inputs include precipitation (Cole et al., 2014; Jeong et al., 2016; Rozos, 2023), wind
direction/speed (Hong and Bhamidimarri, 2012; Cole et al., 2014; Jeong et al., 2016; Kwak et al.,
2016; Temizyurek and Dadaser-Celik, 2018; Abdi et al., 2021; Jiang et al., 2022), barometric pressure
(Cole et al., 2014), landform attributes (Risley et al., 2003; DeWeber and Wagner, 2014; Topp et al.,
2023; Souaissi et al., 2023), and many more (see Table S1).
In the last few years, including the day-of-year as an input, DOY (Qiu et al., 2020; Heddam et
al., 2022; Drainas et al., 2023; Rahmani et al., 2023) and humidity where available (Cole et al., 2014;
Hong and Bhamidimarri, 2012; Kwak et al., 2016; Temizyurek and Dadaser-Celik, 2018; Abdi et al.,
2021), have also shown to better capture the seasonal patterns of SWT (Qiu et al., 2020; Philippus et
al., 2024). With improved access to remote sensing data, there has also been a notable increase of
satellite products such as estimates of sky cover (Cole et al., 2014), solar radiation (Kwak et al., 2016;
Topp et al., 2023; Majerska et al., 2024), sunshine per day (Drainas et al., 2023) and potential ET
(Rozos, 2023; Topp et al., 2023). However, more research is needed to better understand the
influence of newer model inputs on SWT (Zhu and Piotrowski, 2020).
Most recently, SWT studies focused on the CONUS-scale have chosen to use as many model
inputs as available, with Wade et al. (2023), a point-scale CONUS ML study using over 20 variables,
while Rahmani et al. (2023) created a LSTM model and considered over 30 variables to simulate
SWT. Despite the use of diverse data, the models performed only satisfactorily and were deemed not
generalizable, leaving much room for improvement in CONUS-scale modeling of SWT. With the
compilation of larger and larger datasets, feature importance in ML, that is the process of using
techniques to assign a score to model input features based on how good the features are at predicting
a target variable, can be an efficient way to improve data comprehension, model performance, and
model interpretability, the latter of which can dually serve as a transparency marker of which features
are driving predictions. Methods for measuring feature importance include using correlation criteria
(Pearson's r, Spearman's rho), permutation feature importance (shuffling feature values, measuring
decrease in model performance), linear regression feature importance  (larger absolute values indicate
greater importance), or if using CART/RF/gradient boosting, entropy impurity measurements can be
insightful (Venkateswarlu and Anmala, 2023).
*Moved from section 2.3.1, (original lines 246-253) to new section* Model Inputs for ML-SWT:
For example, one technique that can be used to improve ML model parameter selection is the

*Least Absolute Shrinkage and Selection Operator (LASSO),* a regression technique used for feature selection (Tibshirani, 1996). Research utilizing ML models for SWT frequency analysis at ungaged basins used the LASSO method to select explanatory variables for two ML models (Souaissi et al., 2023). The LASSO method consists of a shrinkage process where the method penalizes coefficients of regression variables by minimizing them to zero (Tibshirani, 1996). The number of coefficients set to zero depends on the adjustment parameter, which controls the severity of the penalty. Thus, the method can perform both feature selection and parameter estimation, an advantage when examining large datasets (Xu & Liang, 2021).

**4.** In the second section, the authors do an excellent job summarizing model evaluation metrics. However, considering that ML models are often optimized to achieve superior performance on these metrics, there is (always) a risk of overfitting. Thus, beyond focusing on metrics, the review should also highlight the importance of more rigorous evaluation to further assess generalization ability. For instance, if a SWT model is built to run climate change scenarios, additional testing and more rigorous designs are essential to evaluate the model's ability to generalize over time. For robust long-term predictions, the model is supposed to maintain robust predictive performance in completely unseen periods, rather than being limited to a specific temporal range.

**AUTHOR RESPONSE:** We agree. This comment has similar themes to our response to #3a regarding overfitting and highlighting the need for generalization, please see comment #3a for a full response.

For the comment regarding having ML undergo more rigorous testing, we propose adding the following discussion for more rigorous testing for MLs. We added a few sentences (blue is new) to the Discussion subsection titled "ML as Knowledge Discovery" where we urge for TUURTs (Temporal, Unseen, Ungaged Region Tests)':

Our review finds that ML studies examining SWT have been conducted from a computational perspective, one with a focus on comparing techniques and performance metrics as opposed to explaining the nature of SWT dynamics or influencing processes. While it is understandable that not every ML-SWT paper aims to explain physical processes, we think the SWT community should come together and agree on a baseline of tests that all ML-SWT models should undergo for model robustness and transferability. Along these lines, we urge consideration of TUURTs (temporal, unseen, ungaged region tests) for future ML-SWT models as a helpful step towards not only better modeling practices but also increased model transparency and robustness. For this, we clarify that testing for "unseen" cases means testing only within the developmental dataset, whereas testing for "ungaged" cases means testing for new sites that have not been previously seen by the model at all. Recent ML-SWT studies have only applied one or two of the tests, but not all three (Topp et al., 2023; Hani et al., 2023, Souassi et al., 2023). Siegel et al. (2023), a non-ML SWT paper, tested for ungaged and unseen data but did not perform a temporal test. A relatively new study, Philippus et al. (2024), appears to be the only published SWT-ML study that purposefully applied TUURTs with some success.

*Overall, this review is informative and well-researched, and with more refined organization and deeper exploration of these key issues, it could make a substantial contribution to the field of SWT research.*

**AUTHOR RESPONSE:** Thank you! This would certainly not be possible without the insightful feedback from referees.

---

## Author Comment (AC2)

**Referee #1 Comments**
The manuscript on "*ML in Stream/River Water Temperature Modeling, a review and metrics for*
*evaluation"* focuses on providing a comprehensive review of ML studies, including traditional and recent
methods in ML and AI, on stream temperature modeling and prediction. Overall, the manuscript is well-
written and covers most of the relevant papers, but there are a few strategic points I would like to share
with the authors:
**AUTHOR RESPONSE:** We appreciate the referee's feedback and think the manuscript is much
improved as a result. For reference, we separated some referee comments into a, b, etc., to provide a more
organized response. Thank you for your time and insight. Proposed new/edited text is in BLUE.
**1a.** Figures 1 & 2 & 3 & table 2: The manuscript provides a table for multiple metrics such as R2, NSE,
RMSE, and MAE, and suggested a rate of numbers to rate the ML methods' performances. This table is
based on the metrics that have been achieved by the studies in the previous years which are reflected in
figures 1 & 2 & 3. However, those studies vary in terms of case studies, number of basins included in the
study, running regional or local models. We know that ML models are prone to overfitting, especially for
stream temperature that follows a relatively sinusoidal curve through a year, which means it is more
predictable for complex models such as LSTM. However, it means the models are prone to easily overfit.
Therefore, I suggest the authors encourage the stream temperature researchers to go towards making more
generalizable models and less overfitted. For example, instead of suggesting performance metrics, the
authors can provide a few steps to make sure the models are not overfitted or underfitted. For instance,
always considering a spatial test on ungauged sites (basins). We know that spatial tests are more difficult
tasks rather than temporal tests.
**AUTHOR RESPONSE:** We agree that the SWT studies vary spatially/temporally and that ML models
risk overfitting. We appreciate the referee's comments in pointing out areas of improvement and suggest
adding the following: 1) a new subsection under section 2.4 "SWT Predictions using ML" on
overfitting/underfitting and 2) a diagram showing initial steps to mitigate overfitting. The new text is
below:
Section 2.4.X, Overfitting and Underfitting
When a model is too complex, i.e., has too many features or too many parameters relative to the
number of observations, or is forced to overextend its capabilities, i.e., make predictions with
insufficient training data, the model runs the risk of overfitting (Srivastava et al., 2014). An
overfitting model fits the training data "too well", capturing noise and details that provide high
accuracy on a training dataset, only to perform poorly once the model encounters "unseen" data in
testing/validation (Xu and Liang, 2021). Scenarios where overfitting may be temporarily acceptable
are those where: 1) model development is at its preliminary stages, where the interest is in a "proof of
life" concept, 2) when the objective is to identify heavily-relied on features by the model, i.e., feature
importance, or 3) in highly-controlled modeling environments where the expected data will be
consistently similar to the training dataset. The latter is more likely in certain industrial applications
and unlikely in the changing nature of hydrology.
In contrast, underfitting occurs when a model is too simple to capture any patterns in the data, which
can also lead to terrible performance in training, testing and validation. Underfitting can occur with
inadequate model features, poor model complexity or when regularization techniques, (e.g., L1 or L2
regularization), are over-used, making the model too rigid and unable to respond to changes in the
data. Given the propensity of machine learning models to effectively learn the training data,
underfitting is less of an issue in ML whereas overfitting can be widespread. In the following
diagram, we present an example workflow to transition away from overfitting and towards
generalizability. We further encourage modelers to actively transition towards making more generalizable models, which are in theory, more capable of performing well across diverse scenarios
and datasets, which will become increasingly important with the persistence of climate extremes.

[Figure]

Figure XX. Diagram showing steps that can be taken in modeling process to mitigate overfitting.
**1b.** Therefore, it is acceptable to get lower performance on ungauged basins, however, the metrics should
not be vastly different from temporal tests. A more challenging experiment is to test the trained model on
regions that have not been seen by the model. In theory, if a model has been able to capture true relations
between the driving factors on stream temperature, it should achieve a relatively decent performance on
basins with different hydrologic, geologic, and climatic characteristics from the trained basins. As a
researcher on SWT, I would rather to have a model that passes all these three tests (temporal, ungaged,
unseen regions) with relatively close metrics, rather than having a model that gives high performance in
temporal tests and low performance in the other two tests.
**AUTHOR RESPONSE:** We agree. The referee mentions a key point that having a SWT model pass all
three tests for temporal, ungaged, and unseen regions may be more qualitatively sound, but as of initial
submission, we had not yet seen any ML-SWT papers that test for all three cases. A newly published
example, Philippus et al. (2024), has been added. For example, Topp et al. (2023) held out a region to be
considered "unseen" but did not test for ungaged basins. Hani et al. (2023) used an inverse weighted
distance interpolation method to estimate values for ungaged sites but did not test for "unseen" data.
Souaissi et al. (2023) used a leave-one-out cross-validation technique to mimic the estimation of
quantiles at ungaged sites by temporarily removing the gaged site information, which is arguably not
testing for new, ungaged sites but rather "unseen" (i.e., tested only within the development dataset, not
for new sites). Siegel et al. (2023), a non-ML paper tested for "ungaged" and "unseen" data, but did not
perform a temporal test. We further agree with the theory posited by the referee that a model capturing
true relations should perform acceptably, however, we have yet to see a study that has captured all true
relations.

We have added a few sentences (blue is new) to the Discussion subsection titled "ML as Knowledge Discovery" where we urge for TUURTs (Temporal, Unseen, Ungaged Region Tests)':

Our review finds that ML studies examining SWT have been conducted from a computational perspective, one with a focus on comparing techniques and performance metrics as opposed to explaining the nature of SWT dynamics or influencing processes. While it is understandable that not every ML-SWT paper aims to explain physical processes, we think the SWT community should come together and agree on a baseline of tests that all ML-SWT models should undergo for model robustness and transferability. Along these lines, we urge consideration of TUURTs (temporal, unseen, ungaged region tests) for future ML-SWT models as a helpful step towards not only better modeling practices but also increased model transparency and robustness. For this, we clarify that testing for "unseen" cases means testing only within the developmental dataset, whereas testing for "ungaged" cases means testing for new sites that have not been previously seen by the model at all. Recent ML-SWT studies have only applied one or two of the tests, but not all three (Topp et al., 2023; Hani et al., 2023, Souassi et al., 2023). Siegel et al. (2023), a non-ML SWT paper, tested for ungaged and unseen data but did not perform a temporal test. A relatively new study, Philippus et al. (2024), appears to be the only published SWT-ML study that purposefully applied TUURTs with some success.

**2. Evaluation of Data Requirements:** The manuscript does not extensively discuss the challenges that ML ST modelers are facing with. Different ML models have varying data requirements, but the review does not thoroughly discuss the data needs for each type of model. For example, ML models are dependent on data. If we compare the availability of streamflow observation data availability versus the SWT observation data, we realize there is a massive gap here, which impacts the studies and reduces the SWT model performances. I suggest, while the authors encouraging the researchers and water institutes to collect more data, they add their comments on this issue and discuss how researchers can reduce the impact of this problem in their models.

**AUTHOR RESPONSE:** We agree with the referee that issues remain with data requirement limitations. We propose adding a new 'Discussion' subsection, titled 'ML Data Requirements vs. Availability' stating the following:

While, in recent years, access to hydrologic data has improved (Miller et al., 2022; CUAHSI, 2024), data remains scarce in several hydrologic applications including SWT research, particularly because continual project management and funding to not only place but also maintain stream temperature sensors, can be expensive and/or time-consuming to undertake. As a result, in the 21st century, the scarcity of data remains a large impediment for the application of machine learning in SWT modeling. What is more, the question of data quantity (how much data do you have?) versus quality (how much diverse data is needed?) continues to hinder ML-use in hydrologic applications. Xu and Liang (2021) make the excellent point that one year of streamflow data (can swap for stream temperature) at 15-minute intervals equals about ~35,000 points, which may seem like a lot, but is unlikely to be enough to properly train a ML model due to autocorrelation and limited exposure to diverse types of data that are naturally encountered with a longer time-series (Xu and Liang, 2021). For example, machine learning models may only predict flood volumes they have previously seen (Kratzert et al., 2019). While data requirements for ML remain high, there are some strategies that researchers have used to alleviate the impact of this issue.

One strategy that hydrologists in other fields have used to tackle this problem is data augmentation, which can be applied spatially or temporally to create new training examples that the ML model can learn from. Spatial augmentation can be done by means of interpolation methods, i.e., kriging or distance weighting to create new data points or by generating synthetic data based on expected physical patterns to fill gaps in data coverage (Baydaroğlu and Demir, 2024). Temporal data augmentation can be done by shifting, scaling or adding noise to existing time series to create new training examples for the model to consider (Skoulikaris et al., 2022). Alternatively, and not a new idea, would be to use the statistical technique known as seasonal decomposition, which breaks down a time series into its main components, i.e., the trend, seasonal patterns and residual components (Apaydin et al., 2021; He et al., 2022). These can then be recombined to generate new data and train the model for improved accuracy (Apaydin et al., 2021). In addition to data augmentations, data requirements can be alleviated by considering the help of unsupervised transfer learning, i.e., use pre-trained models on similar tasks to reduce amount of data needed for training, or semi-supervised learning, such as few shot learning, i.e., combine a small percent of labeled data with larger percent of unlabeled data to improve model performance (Yang et al., 2023). By implementing these strategies, researchers in other hydrologic fields have shown that models can be improved with less data, strategies that are likely transferable to SWT research.

**3. Future Directions Could Be Expanded:** Although the paper concludes with a general discussion of future challenges, it does not offer specific, actionable directions for future research. Highlighting key areas where ML can advance, such as the use of satellite data, sensor networks, or the fusion of climate models with ML, would provide more meaningful insights. In this concept, we can learn from hydrologic community and capitalize on their experience and what they learned. The ML hydrologic community is moving toward making global models, incorporating mechanistic models into their ML framework and learning the governing factors, flow prediction with predicted inputs (predicted meteorological inputs) and last but not least, providing a seamless simulation in streams in CONUS/global scale. Therefore, I would ask the authors to add their comments on where the future direction of SWT community should be and how SWT community can achieve the future objectives and what the barriers are.

**AUTHOR RESPONSE:** We agree and appreciate the referee's feedback. We propose adding a new 'Discussion' subsection, titled '4.3 Future Directions of SWT Modeling', with the following:

> The utility of ML in hydrologic modeling has come a long way, with interest seemingly growing exponentially (Nearing et al., 2021). With the novelty of ML, it is easy to get lost in the value of how well a model performs and ignore the science, but with several decades of ML-experience, we think it necessary to urge the scientific community to purposefully use ML address physically-meaningful questions and not just create ML for the sake of creating. Given this, Varadharajan et al. (2022) laid out an excellent discussion on opportunities for advancement of ML in water quality modeling, see section 3 of publication (Varadharajan et al., 2022). Here we highlight some of the questions from Varadharajan et al. (2022) that can be considered in the context of what the objectives of the SWT community should be in the ML era, namely: 1) How do we use physical knowledge (re: heat exchange equations, radiation influence) to improve models and process understanding? Rahmani et al. (2023) coupled NNs with the physical knowledge from SNTEMP, a one-dimensional stream temperature model that calculates the transfer of energy to or from a stream segment by either heat flux equations or advection, but found that even with SNTEMP, their flexible NNs exhibited substantial variance in prediction and needed to be constrained by further multi-dimensional assessments (Rahmani et al., 2023). In short, if our use of physics in machine learning makes our models worse, we must know why.
>
> A second question that needs addressing is 2) How do we deal with predictive uncertainty in ML used for SWT modeling? According to Moriasi et al. (2007), uncertainty analysis is the process of quantifying the level of confidence in any given model output based on five guidelines: 1) the quality and amount of observations (data), 2) the lack of observations due to poor or limited field monitoring, 3) the lack of knowledge of physical processes or operational procedures (instrumentation), 4) the approximation of our mathematical equations, and 5) the robustness of model sensitivity analysis and calibration. For example, in rainfall-runoff modeling, researchers have proposed benchmarking to examine uncertainty predictions of ML rainfall-runoff modeling (Klotz et al., 2022). For stream temperature modeling, researchers have attempted to address the role of uncertainty in deep learning model (RGCN, LSTM) prediction using the Monte Carlo Dropout (Zwart, Oliver, et al., 2023) and a unimodal mixture density network approach (Zwart, Diaz, et al., 2023).

Other questions that SWT-ML studies should consider is 3) How do we make ML models generalize better, specifically with regards to ungaged basins? And 4) How can ML models be improved to predict extremes? As ML models advance to use satellite data, include more sensor networks and/or couple with climate models, there is a logical next step toward creating generalizable models that can account for extremes. In our review, only two papers by the same group (Rahmani et al., 2020, 2023) conducted a CONUS-scale approach towards SWT-ML modeling, omitting hydrologically important regions in the southwest (CA) and southeast (FL). Recently, a satellite remote sensing paper used RF to model monthly stream temperature across the CONUS and tested for temporal (walk-forward validation), unseen and 'true' ungaged regions (Philippus et al., 2024). We have also learned that ML models such as LSTMs, generally only make predictions within the bounds of their training data (Kratzert et al., 2019), which is a limitation for predicting extremes. Thus, we strongly urge the community to work towards ML models that generalize better and/or are more robust towards predictions of extremes.

Finally, 5) How can we build ML models such that they are seen as trustworthy and interpretable by the hydrologic community? To answer this question, we must address a technical barrier (black-box issues, data limitations, model uncertainty) and a social barrier (i.e., educated skepticism of ML due to novelty, little understanding of computer science basics and/or coding experience). If we are to incorporate ML into more of the decision-making process, it makes sense that ML must be transparent and understandable to more than just computer scientists (Varadharajan et al., 2022). For example, Topp et al. (2023) recently used explainable AI to elucidate how ML architectures affected the SWT model's spatial and temporal dependencies, and how that in turn affected the model's accuracy. Addressing this technical barrier can also be done by improving access to data, which has seen remarkable progress thanks to web repositories such as NSF-funded CUAHSI's Hydro share (CUAHSI, 2024) and GitHub (GitHub, 2024). In the United States, data access to state and locally-based data remains limited, and should be addressed. In terms of the social barrier, education about ML and ML-use is key. Societal interest in ML has thankfully also lead to a plethora of educational resources and ML walk-through videos and tutorials in Tensorflow (Abadi et al., 2015), PyTorch (Abadi et al., 2015), and Google Colab (Bison, 2019). With how fast ML-use is evolving, short communication pieces (Lapuschkin et al., 2019) and opinion pieces (Kratzert et al., 2024) with clear examples about an ML-issue and practical solutions could also help make ML challenges more transparent and therefore accessible to the hydrologic community-at-large.

**Added citations used for new subsection, 4.3 Future Directions of SWT Modeling:**

1) Apaydin, H., Taghi Sattari, M., Falsafian, K., and Prasad, R.: Artificial intelligence modelling integrated with Singular Spectral analysis and Seasonal-Trend decomposition using Loess approaches for streamflow predictions, Journal of Hydrology, 600, 126506, https://doi.org/10.1016/j.jhydrol.2021.126506, 2021.

2) Baydaroğlu, Ö. and Demir, I.: Temporal and spatial satellite data augmentation for deep learning-based rainfall nowcasting, Journal of Hydroinformatics, 26, 589–607, https://doi.org/10.2166/hydro.2024.235, 2024.

3) CUAHSI. 2024. Consortium of Universities for the Advancement of Hydrologic Science, Inc. (CUAHSI) Water Data Portal: https://www.cuahsi.org/community/water-data-portals, last access: 13 November 2024.

4) Kratzert, F., Gauch, M., Klotz, D. and Nearing, G., 2024. HESS Opinions: Never train an LSTM on a single basin. Hydrology and Earth System Sciences Discussions, 2024, pp.1-19.

5) Kwak, J., St-Hilaire, A., and Chebana, F.: A comparative study for water temperature modelling in a small basin, the Fourchue River, Quebec, Canada, Hydrological Sciences Journal, 1–12, https://doi.org/10.1080/02626667.2016.1174334, 2016.

6) Philippus, D., Sytsma, A., Rust, A., and Hogue, T. S.: A machine learning model for estimating the temperature of small rivers using satellite-based spatial data, Remote Sensing of Environment, 311, 114271, https://doi.org/10.1016/j.rse.2024.114271, 2024.

7) Nearing, G. S., Kratzert, F., Sampson, A. K., Pelissier, C. S., Klotz, D., Frame, J. M., Prieto, C., and Gupta, H. V.: What Role Does Hydrological Science Play in the Age of Machine Learning?, Water Resources Research, 57, e2020WR028091, https://doi.org/10.1029/2020WR028091, 2021.

8) Skoulikaris, C., Venetsanou, P., Lazoglou, G., Anagnostopoulou, C., and Voudouris, K.: Spatio-Temporal Interpolation and Bias Correction Ordering Analysis for Hydrological Simulations: An Assessment on a Mountainous River Basin, Water, 14, 660, https://doi.org/10.3390/w14040660, 2022.

9) Srivastava, N., Hinton, G., Krizhevsky, A., Sutskever, I., and Salakhutdinov, R.: Dropout: A Simple Way to Prevent Neural Networks from Overfitting, Journal of Machine Learning Research, 15, 30, 2014.

10) Yang, M., Yang, Q., Shao, J., Wang, G., and Zhang, W.: A new few-shot learning model for runoff prediction: Demonstration in two data scarce regions, Environmental Modelling & Software, 162, 105659, https://doi.org/10.1016/j.envsoft.2023.105659, 2023.

11) GitHub. 2024. About Git and Github: https://docs.github.com/en/get-started/start-your-journey/about-github-and-git, last access: 14 November 2024.

12) Lapuschkin, S., Wäldchen, S., Binder, A., Montavon, G., Samek, W. and Müller, K.R., 2019. Unmasking Clever Hans predictors and assessing what machines really learn. Nature communications, 10(1), p.1096.

13) Zwart, J.A., Oliver, S.K., Watkins, W.D., Sadler, J.M., Appling, A.P., Corson-Dosch, H.R., Jia, X., Kumar, V. and Read, J.S., 2023. Near-term forecasts of stream temperature using deep learning and data assimilation in support of management decisions. JAWRA Journal of the American Water Resources Association, 59(2), pp.317-337.

14) Zwart, J.A., Diaz, J., Hamshaw, S., Oliver, S., Ross, J.C., Sleckman, M., Appling, A.P., Corson-Dosch, H., Jia, X., Read, J. and Sadler, J., 2023. Evaluating deep learning architecture and data assimilation for improving water temperature forecasts at unmonitored locations. *Frontiers in Water*, 5, p.1184992.

15) Klotz, D., Kratzert, F., Gauch, M., Keefe Sampson, A., Brandstetter, J., Klambauer, G., Hochreiter, S. and Nearing, G., 2022. Uncertainty estimation with deep learning for rainfall–runoff modeling. Hydrology and Earth System Sciences, 26(6), pp.1673-1693.

16) M. Abadi, A. Agarwal, P. Barham, E. Brevdo, Z. Chen, C. Citro, G. S. Corrado, A. Davis, J. Dean, M. Devin, S. Ghemawat, I. Goodfellow, A. Harp, G. Irving, M. Isard, R. Jozefowicz, Y. Jia, L. Kaiser, M. Kudlur, J. Levenberg, D. Mané, M. Schuster, R. Monga, S. Moore, D. Murray, C. Olah, J. Shlens, B. Steiner, I. Sutskever, K. Talwar, P. Tucker, V. Vanhoucke, V. Vasudevan, F. Viégas, O. Vinyals, P. Warden, M. Wattenberg, M. Wicke, Y. Yu, and X. Zheng. TensorFlow: Large-scale machine learning on heterogeneous systems. 2015. TensorFlow. Website: https://www.tensorflow.org/

17) A. Paszke, S. Gross, F. Massa, A. Lerer, J. Bradbury, G. Chanan, T. Killeen, Z. Lin, N. Gimelshein, L. Antiga, A. Desmaison, A. Köpf, E. Yang, Z. DeVito, M. Raison, A. Tejani, S. Chilamkurthy, B. Steiner, L. Fang, J. Bai, S. Chintala. 2019. PyTorch: An Imperative Style, High-Performance Deep Learning Library. Website: https://arxiv.org/abs/1912.01703

18) Bisong, E. (2019). Google Colaboratory. In: Building Machine Learning and Deep Learning Models on Google Cloud Platform. Apress, Berkeley, CA. Website: https://doi.org/10.1007/978-1-4842-4470-8_7

**4.** The manuscript walked through many ML and AI models. An important factor of the ML and AI models are the inputs. I assume you faced a variety of inputs that have been used in the models. That would be informative to the readers, if the authors add their observations that what kind of inputs that have been missed to be used, either because it is not available yet or it is even missed. For instance, whether there is any geophysical attribute, climatic attributes, or any forcings that is worth to be extracted and used in ML models.

**AUTHOR RESPONSE:** We appreciate the referee's feedback. In the Supplementary Materials, Table S1 contains some of the suggested data by the referee, such as: period considered, region examined, temporal resolution of SWT, spatial scale of study, and hydrometeorological parameters used for modeling. We provided the information as Supplementary Material because Tables S1 and S2 are seven pages alone, which may risk making the review lengthier than it already is. We have added text to the manuscript regarding model inputs and moved the LASSO paragraph (original lines 247-253) to this section because
we think it can more smoothly follow the paragraph on feature importance.
This section will precede the "Local" and "Regional" subsections of 2.4 and be titled Model Inputs for
ML-SWT:
Using air temperature (AT) to better understand SWT has been considered since at least the
1960s, when Ward (1963) and Edinger et al. (1968) discussed the influence of air temperature on
SWT. Since then, studies have used varying input variables (see Table S1), however, the model inputs
of AT and SWT continue to be the most used in ML-modeling studies. In particular, studies have
used AT from time periods outside of the known SWT record to improve model performance (Sahoo
et al., 2009; Piotrowski et al., 2015; Graf et al., 2019). In addition to AT and SWT, flow discharge has
been used to attempt to constrain SWT (Foreman et al., 2001; Tao et al., 2008; St-Hilaire et al., 2011;
Grbić et al., 2013; Piotrowski et al., 2015; Graf et al., 2019; Qiu et al., 2020). Traditionally-used
model inputs include precipitation (Cole et al., 2014; Jeong et al., 2016; Rozos, 2023), wind
direction/speed (Hong and Bhamidimarri, 2012; Cole et al., 2014; Jeong et al., 2016; Kwak et al.,
2016; Temizyurek and Dadaser-Celik, 2018; Abdi et al., 2021; Jiang et al., 2022), barometric pressure
(Cole et al., 2014), landform attributes (Risley et al., 2003; DeWeber and Wagner, 2014; Topp et al.,
2023; Souaissi et al., 2023), and many more (see Table S1).
In the last few years, including the day-of-year as an input, DOY (Qiu et al., 2020; Heddam et
al., 2022; Drainas et al., 2023; Rahmani et al., 2023) and humidity where available (Cole et al., 2014;
Hong and Bhamidimarri, 2012; Kwak et al., 2016; Temizyurek and Dadaser-Celik, 2018; Abdi et al.,
2021), have also shown to better capture the seasonal patterns of SWT (Qiu et al., 2020; Philippus et
al., 2024). With improved access to remote sensing data, there has also been a notable increase of
satellite products such as estimates of sky cover (Cole et al., 2014), solar radiation (Kwak et al., 2016;
Topp et al., 2023; Majerska et al., 2024), sunshine per day (Drainas et al., 2023) and potential ET
(Rozos, 2023; Topp et al., 2023). However, more research is needed to better understand the
influence of newer model inputs on SWT (Zhu and Piotrowski, 2020).
Most recently, SWT studies focused on the CONUS-scale have chosen to use as many model
inputs as available, with Wade et al. (2023), a point-scale CONUS ML study using over 20 variables,
while Rahmani et al. (2023) created a LSTM model and considered over 30 variables to simulate
SWT. Despite the use of diverse data, the models performed only satisfactorily and were deemed not
generalizable, leaving much room for improvement in CONUS-scale modeling of SWT. With the
compilation of larger and larger datasets, feature importance in ML, that is the process of using
techniques to assign a score to model input features based on how good the features are at predicting
a target variable, can be an efficient way to improve data comprehension, model performance, and
model interpretability, the latter of which can dually serve as a transparency marker of which features
are driving predictions. Methods for measuring feature importance include using correlation criteria
(Pearson's r, Spearman's rho), permutation feature importance (shuffling feature values, measuring
decrease in model performance), linear regression feature importance  (larger absolute values indicate
greater importance), or if using CART/RF/gradient boosting, entropy impurity measurements can be
insightful (Venkateswarlu and Anmala, 2023).
*Moved from section 2.3.1, original lines 246-253 to new section* Model Inputs for ML-SWT:
For example, one technique that can be used to improve ML model parameter selection is the
*Least Absolute Shrinkage and Selection Operator (LASSO),* a regression technique used for feature
selection (Tibshirani, 1996). Research utilizing ML models for SWT frequency analysis at ungaged
basins used the LASSO method to select explanatory variables for two ML models (Souaissi et al.,
2023). The LASSO method consists of a shrinkage process where the method penalizes coefficients
of regression variables by minimizing them to zero (Tibshirani, 1996). The number of coefficients set
to zero depends on the adjustment parameter, which controls the severity of the penalty. Thus, the
method can perform both feature selection and parameter estimation, an advantage when examining
large datasets (Xu & Liang, 2021).

**5. Lack of Clear Structure in the Evaluation:** Although the paper aims to summarize the performance
evaluation metrics for ML models in SWT prediction, the organization of these sections feels somewhat
scattered. A more systematic approach could improve clarity, such as separating the analysis based on
time scales (e.g., hourly, daily, monthly) or spatial scales (local, regional, continental). This would make
it easier for readers to find the relevant insights based on their application. For instance, a stream
temperature model in monthly scale is different from a daily or hourly scale models on many aspects. As
an example, the complexity of a daily model is different from a monthly temperature models. A monthly
model may not need all inputs of a daily model to capture the monthly changes. The authors can add their
overall opinion of what types of models are better fitted to which time scale. In ML models, it is
important to know the scope of the model, whether it is a local model that needs to be calibrated site by
site, or it is a model that  is designed to work for multiple sites (a regional model). I believe that would be
informative to consider the modeling approach when methods are compared.
**AUTHOR RESPONSE:** We appreciate the opportunity to clarify. Initially, we did create a performance
metric comparison by spatial scale for the most-cited metric, RMSE (42 papers cited) and plotted RMSE
by study for regional/CONUS scale and local scale (found in the HYDROSHARE repository but not in
the manuscript), however we found minimal performance metric differences between the
regional/CONUS studies and the local scale studies, which we state on Table 1 in the manuscript.
One of the possible reasons why we found no differences is due to the inherent variability of each
individual publication's goals and a self-limiting behavior where earlier studies published less data than
later studies. Additionally, one of the challenges of performance metric choice in the hydrologic modeling
community is that authors choose whatever performance metric they prefer with little regard to what
everyone else is doing (though this is improving). This inconsistency in choice limits which performance
metrics we can summarize for readers and makes for challenging cross-comparison. Given how fast ML
is advancing and being applied for hydrologic applications, we do not believe it wise to opinionate on
which ML model is better or worse. We think the choice is in the reader's hands and comes down to what
the research question/goal is, the time frame of the research project, and the author's own objectives. We
think that what we have done instead, with summarizing publications (see Tables S1 and S2) and
highlighting performance metrics, allows the reader to identify what has already been done in the ML-
SWT field so that they can then make their own informed decisions about their research questions and
methods. As part of the supplementary info, Tables S1 includes summarized information stating the time
scale, spatial scale, region and time period considered of each study while Table S2 lists the data analysis
techniques and/or ML algorithms used, as well as the training/validation/testing percentages/time periods
as reported by the study.

[Figure]

[Figure]

**6.** The authors need to decide first who are the readers of the papers. Whether the paper serves to new-commers to ML and AI methodologies in stream temperature community or it serves to researchers that are already familiar with basics of ML and AI methods.

**AUTHOR RESPONSE:** We agree with the referee that the purpose of the review should be more clearly stated. We drafted this paper to serve as a middle ground between traditional modelers and more well-versed ML users. The intended audience are hydrologic modelers who have heard of AI/ML and want a summary of what has been done in SWT modeling using ML. Our dual objective is also for this to be a reference for what to expect from ML performance. At the same time, we want ML researchers to be aware of where their models stand compared to other modelers while communicating that an "A+ grade" is actually more common (and therefore the new average) relative to what they are used to in hydrologic modeling. We have added a few sentences in the introduction, under section 1.2 'Study Objectives' of the manuscript to state who the intended audience is:

**1.2. Study Objective (new in blue)**

*The current work includes an extensive literature review of studies that used ML algorithms/models for river/SWT modeling, hindcasting and forecasting.* The intent of this review is two-fold: 1) to introduce ML for hydrologists who have computer modeling experience and are interested in pursuing ML-use for their SWT studies, and 2) to provide a broad overview of machine learning applications in SWT. For ML experts, we think that this review could also prove useful as reference for how ML has been applied in the field of SWT modeling and where improvement is needed. Overall, this article aims to serve as a bridge between hydrologists and machine learning experts. *Our review includes papers cited by Zhu and Piotrowski (2020), who previously conducted a study of ANNs used in SWT modeling, however, we provide a comprehensive examination of peer-reviewed journals that use any type of artificial intelligence/ML algorithm to model or evaluate river/SWT [...]*

**7a.** While the paper provides an extensive review of ML applications in SWT modeling, it focuses heavily on listing the types of ML models used rather than deeply analyzing their applications, strengths, weaknesses, and performance differences. A more critical analysis of the pros & cons of each model type could provide greater value to researchers choosing the appropriate model for their specific needs. To provide a few examples, I refer to lines 136 – 143 & lines 146 – 159 & lines 263 - 292.

**AUTHOR RESPONSE**: Thank you for the opportunity to clarify. We provided supplementary tables to summarize study information, for example, Tables S1 includes summarized information stating the time scale, spatial scale, region and time period considered of each study while Table S2 lists the data analysis techniques and/or ML algorithms used, as well as the training/validation/testing percentages/time periods as reported by the study. We think the "pros/cons" and "strengths/weakness" vary depending on the research goal and question, and the robustness of ML models allows them to cater to most problems, which is why we think instead of opinionating, it is better that we provide concrete specifications on the models used and allow the reader to decide based on their objectives.

**7b.** The first half of the paragraph that is written in lines 136 – 143 explains the fundamentals of the method, which may not be necessary to be long, and the rest is an example of the method usage. However, this paragraph could have been enriched by statements like the advantages and disadvantages of this method compared to other existing ML methods or even to a linear regression method, or a 1D mechanistic method (although they are not ML methods, but the comparison is beneficial to the readers). The authors also can add their statement of under what conditions they think the method is beneficial.

**AUTHOR RESPONSE**: We agree and show how we could edit the text to include describing the advantages and disadvantages of K-nn:

K-nearest neighbors (K-nn) is a  versatile supervised ML algorithm (Fix & Hodges, 1952; Cover & Hart, 1967) used to solve nonparametric classification and regression problems.  The K-nn algorithm uses proximity between data points to make classifications or evaluations about the grouping of any given data point (Acito, 2023). K-nn gained popularity in the 2010s due to its simplicity in implementation and understanding, making it accessible to hydrologic researchers and practitioners.  For example, St.-Hilaire et al. (2011) used various K-nn model configurations to model SWT for the Moisie River in northern Quebec, Canada, finding that the best K-nn model required prior-day SWT data and day-of-year (DOY), an indicator of seasonality . Other advantages of K-nn include its non-assumptions of the underlying distribution of the data, allowing it to handle nonlinear complexities without requiring a solid model structure as is the case for some physical models (St-Hilaire et al., 2011). The disadvantages of K-nn are quite large however, as it has been found to be computationally intensive, requiring extensive cross-validation, is affected by irrelevant/redundant features that impact performance, and is impractical for large-scale applications (i.e., scalability issues), due to its high memory and computational requirements (Acito, 2023). For example, Heddam et al. (2022)  compared K-nn with other ML algorithms, finding that K-nn was outperformed by other MLs such as least squares support vector machine and neural networks.  The use of K-nn may still be apt for simple, local cases but we advise considering other MLs for more complex or larger-use cases due to the aforementioned.

**7c. Lines 146 – 153** explains PCA & ck-means clustering on data reduction application, however, it is not clear under what conditions we can use them.

**AUTHOR RESPONSE**: We agree. We propose adding text to clarify:

Krishnaraj and Deka (2020) used *K-means* to organize spatial grouping for water quality monitoring stations for dry and wet regions along the Gangas River basin in India to identify whether pollution patterns could be discerned.

Using *PCA*, Krishnaraj and Deka (2020) found that certain water quality parameters (dissolved oxygen, sulfate, electrical conductivity) were more dominant in the dry season compared to the wet season (total dissolved solids, sodium, potassium, sodium, chlorine, chemical oxygen demand), data which could be used to cater the monitoring program to the important parameters. In their study, SWT was not a dominant parameter, likely in part because the SWT of large downstream rivers like the Gangas River are generally less variable due to their larger volume and stronger thermal buffer.

**7d.** Additionally, that would be nice for readers if the authors add feature importance to their comparison as it has been used more frequently in streamflow and soil moisture prediction studies.

**AUTHOR RESPONSE**: We agree and added text on feature importance to a section on model inputs as suggested (please see comment #4 for full text). The text specific to feature importance is below:

Most recently, SWT studies focused on the CONUS-scale have chosen to use as many model inputs as available, with Wade et al. (2023), a point-scale CONUS ML study using over 20 variables, while Rahmani et al. (2023) created a LSTM model and considered over 30 variables to simulate SWT. Despite the use of diverse data, the models performed only satisfactorily and were deemed not generalizable, leaving much room for improvement in CONUS-scale modeling of SWT. With the compilation of larger and larger datasets, feature importance in ML, that is the process of using techniques to assign a score to model input features based on how good the features are at predicting a target variable, can be an efficient way to improve data comprehension, model performance, and model interpretability, the latter of which can dually serve as a transparency marker of which features are driving predictions. Methods for measuring feature importance include using correlation criteria (Pearson's r, Spearman's rho), permutation feature importance (shuffling feature values, measuring decrease in model performance), linear regression feature importance (larger absolute values indicate greater importance), or if using CART/RF/gradient boosting, entropy impurity measurements can be insightful (Venkateswarlu and Anmala, 2023).

**7e.** Lines 263 – 292 are organized in three paragraphs while providing general knowledge about ANNs with relatively less direct relations to water temperature application.

**AUTHOR RESPONSE**: We appreciate the reviewer's feedback and are open to making changes to improve the manuscript for the reader. Referee #3 made a similar comment about this section, and we now wonder if it would be better to provide the description of ANN variants and alternatives (lines 263-320) as part of an appendix. We think it would still be helpful to keep the information, but we also agree that it may be too extensive for the main text. In this way, the manuscript can be made more concise while also keeping the details as a section of the manuscript for anyone who is interested in reading further.

Following this line of thinking, we can add the following to point the reader to the appendix:

"For more detail on traditional ANNs, with descriptions of ANN variants and backpropagation alternatives, we refer the reader to appendix A."

**Minor corrections:**

1. Line 13: There is a typo that changes the meaning of the sentence. It should be "… with in situ …" or "… with in-situ …".

**AUTHOR RESPONSE:** Thank you for pointing this out, we have fixed the typo to read "with in-situ".

2. Line 132: There is a typo here too. It should be "long short-term memory". Although I am trying to catch them, there is a chance that I miss some of them. I recommend the authors to carefully re-read the manuscript or ask help from a fresh pair of eyes to find these types of typos.

**AUTHOR RESPONSE:** Thank you! We have revised the text to read "long short-term memory" and reviewed the text accordingly.

3. Lines 208 – 210: to make the sentence more accurate, it needs to be stated whether these are local models or one model for multiple sites. Additionally, I believe by "NNs" here, the authors mean feedforward neural network, which are totally different from recurrent neural networks.

**AUTHOR RESPONSE:** Yes, we agree with both points. We have clarified that a feed-forward NN was used and revised the sentence to make it more accurate:

 A SWT modeling study comparing the output of three model versions of DT, GPR, and feed-forward neural networks for  multiple sites , found that DTs  could perform similarly to GPR and feed-forward neural networks when detailed statistics of air temperature, day-of-year, and discharge were included  (Zhu, Nyarko, Hadzima-Nyarko, Heddam, et al., 2019).

4. Line 541: "at" is missed. It is .. All journals examined used at least …"

**AUTHOR RESPONSE:** Thank you! We have added the word "at".

---

## Author Comment (AC3)

**Referee #3 Comments**

I believe that this manuscript is a very useful and extensive methods literature review regarding stream temperature modeling. I would recommend approval with minor revisions to provide additional details from the reviewed literature and correct minor writing aspects; I had no problem with the general structure/flow or quality.

**AUTHOR RESPONSE**: We thank the referee for their time and feedback, we believe the manuscript is better as a result. We address specific referee comments below. Proposed new/edited text is in BLUE.

**1.** Section 2.3.3 ("Newer/recent ML algorithms") introduces RNNs, CNNs, and GNNs sufficiently, but it should probably give some description and reference to attention-based transformers. I am not aware of their application to SWT, but they are responsible for broader interest in ML (e.g., ChatGPT, which was cited earlier) and have had mixed success in hydrologic modeling. This class of models seems easily placed as a future direction.

**AUTHOR RESPONSE**: We agree. A literature search on Google Scholar in November 2024 found no publications specifically using attention-based transformers for SWT, but we are happy to add some text about their potential to section 2.3.3:

Attention-based transformers are a more novel type of deep learning that has led to advancements in natural language processing, in the form of ChatGPT, Microsoft's CoPilot, Google's Gemini and others. Due to their exponential success in the last few years, attention-based transformer models have been used in geological science fields such as oceanography for sea surface temperature prediction (Shi et al., 2024), hydrology for streamflow and runoff prediction (Ghobadi and Kang, 2022; Wei, 2023) and remote sensing for streambed land use change classification (Bansal and Tripathi, 2024). As a relatively new DL tool, attention-based transformers have yet to be used for SWT, but their aforementioned applications in other geological science fields suggest it is only a matter of time before we see their use in SWT modeling.

**2.** There are some examples of unusual subsection and paragraph formatting. For example, section 1.1 is one paragraph which is approximately 1 page long. It seems that this is excessively large for one paragraph and that a named subsection should perhaps be more than just one (regularly sized) paragraph. Line 201 has another approximately 1-page-long paragraph, this area might be better organized with another level of subsections rather than fitting the more extensive references of decision trees into 1 paragraph.

**AUTHOR RESPONSE**: We appreciate the opportunity to clarify. For section 1.1 (line 35), the 2$^{nd}$ paragraph begins on line 46, with the words "*Aided by the continued...*". The same occurs after Line 201, where the RF and XGBoost paragraph begins on line 238. The manuscript follows the Copernicus manuscript template (screenshot below) which appears to not provide for paragraph indentation.

**1 Section (as Heading 1)**

Suspendisse a elit ut leo pharetra cursus sed quis diam (Smith et al., 2014; Miller and Carter, 2015). Nullam dapibus, ante vitae congue egestas, sem ex semper orci, vel sodales sapien nibh sed lectus. Etiam vehicula lectus quis orci ultricies dapibus. In sit amet lorem egestas, pretium sem sed, tempus lorem.

**1.1 Subsection (as Heading 2)**

Quisque cursus massa sed urna congue, ac convallis neque consectetur. Proin faucibus neque non metus mollis, suscipit pretium nisl blandit. In hac habitasse platea dictumst.

At the referee's suggestion, we can add subsections to section 2.3.1 to distinguish algorithms as follows:

        2.3.1.1 K-nearest neighbors (starts line 138),
        2.3.1.2 Cluster analysis and variants (line 145),
        2.3.1.3 Support vector machine and regression (line 160),
        2.3.1.4 Gaussian Process Regression (line 189),
        2.3.1.5 Decision trees and Classification and Regression Trees (line 202),
        2.3.1.6 Random Forests and XGBoost (line 215)

We also think that we can make section 2.3.1.6 (lines 226-253) more concise now because model inputs are now a separate section (section 2.4.X). Below is our suggested reduction, with the last LASSO paragraph also being moved to model inputs and selection:

 RF and XGBoost have been used to predict SWT for Austrian catchments with minor differences in model performance,  with a median RMSE difference of 0.08 °C between tested ML models (Feigl et al., 2021). Using RF and XGBoost along with four other ML models, Jiang et al. (2022)  estimated daily SWT below dams in China, finding day of year, stream flow flux and AT to be  most influential for the prediction of SWT (Jiang et al., 2022). Weierbach et al. (2022) used XGBoost and SVR to predict SWT at monthly time scales for the Pacific Northwest region of the U.S., finding that an ensemble XGBoost outperformed all modeling configurations for spatiotemporal predictions in unmonitored basins.  AT as the primary driver of monthly SWT Zanoni et al. (2022) used RF and a deep learning model to develop regional models of SWT and other water quality parameters, finding that RF performance was comparatively less effective at detecting non-linear relationships, though both models identified AT as most influential  (Zanoni et al., 2022).

Souassi et al. (2023)  compared the performance of  RF and XGBoost, with non-parametric models for the regional estimation of maximum SWT at ungaged locations in Switzerland, finding no significant differences between the ML  and  non-parametric model performances, which was attributed to the lack of a large dataset . Hani et al. (2023) used four supervised ML models – MARS, GAM, SVM, and RF to model potential thermal refuge area (PTRA) at an hourly timestep for two tributary confluences of the Sainte-Marguerite River in Canada. RF had the highest accuracy at both locations in terms of hourly PTRA estimates and modeling SWT (Hani et al., 2023). Wade et al. (2023) conducted a CONUS-scale study using RF  with four years of daily SWT and discharge to examine maximum SWT. found that  Study findings identified AT as most influential control followed by other properties (watershed characteristics, hydrology, anthropogenic impact).

**3.** There is an extensive background of traditional ANNs (2.3.2) which is debatably too extensive given the description of ANN variants and backpropagation alternatives (e.g., lines 284-320), which are relatively niche and rare. The content already exists and is not wrong, but if length were a concern, I would reduce this area.

**AUTHOR RESPONSE**: We appreciate the reviewer's feedback and are open to making changes to improve the manuscript for readability. Referee #1 made a similar comment about this section, and we now propose providing the description of ANN variants and alternatives (lines 263-320) as part of an appendix. We think it would still be helpful to keep the ANN information, but we also agree that it may be too extensive for the main text. In this way, the manuscript can be made more concise while also keeping the details as a section of the manuscript for anyone who is interested in reading further. Following this line of thinking, we can add the following to point the reader to the appendix:

> "For more detail on traditional ANNs, with descriptions of ANN variants and backpropagation alternatives, we refer the reader to appendix A."

**4.** This work does not address predictive uncertainty, or the lack thereof associated with the ML literature review. I think that would be a worthwhile addition because I suspect most efforts lack that (e.g., referring to https://doi.org/10.5194/hess-26-1673-2022 ). A counterexample to the lack of uncertainty quantification, which may also be relevant to section 2.5, could be work led by Jacob Zwart focusing on SWT for reservoir operations (thermal releases). Examples being https://doi.org/10.1111/1752-1688.13093 or https://doi.org/10.3389/frwa.2023.1184992

**AUTHOR RESPONSE**: We appreciate the referee's insight in bringing these publications to our attention. Based on their relevancy, we have added Klotz et al. 2022, Zwart et al. 2023a and 2023b to our manuscript and included their RMSE values in our review. First, we added text on predictive uncertainty in the new 'Discussion' subsection, titled 'Future Directions of SWT Modeling' (this section also addresses ref #1, comment #3):

> The utility of ML in hydrologic modeling has come a long way, with interest seemingly growing exponentially (Nearing et al., 2021). With the novelty of ML, it is easy to get lost in the value of how well a model performs and ignore the science, but with several decades of ML-experience, we think it necessary to urge the scientific community to purposefully use ML address physically-meaningful questions and not just create ML for the sake of creating. Given this, Varadharajan et al. (2022) laid out an excellent discussion on opportunities for advancement of ML in water quality modeling, see section 3 of publication (Varadharajan et al., 2022). Here we highlight some of the questions from Varadharajan et al. (2022) that can be considered in the context of what the objectives of the SWT community should be in the ML era, namely: 1) How do we use physical knowledge (re: heat exchange equations, radiation influence) to improve models and process understanding? Rahmani et al. (2023) coupled NNs with the physical knowledge from SNTEMP, a one-dimensional stream temperature model that calculates the transfer of energy to or from a stream segment by either heat flux equations or advection, but found that even with SNTEMP, their flexible NNs exhibited substantial variance in prediction and needed to be constrained by further multi-dimensional assessments (Rahmani et al., 2023). In short, if our use of physics in machine learning makes our models worse, we must know why.
>
> A second question that needs addressing is 2) How do we deal with predictive uncertainty in ML used for SWT modeling? According to Moriasi et al. (2007), uncertainty analysis is the process of quantifying the level of confidence in any given model output based on five guidelines: 1) the quality and amount of observations (data), 2) the lack of observations due to poor or limited field monitoring, 3) the lack of knowledge of physical processes or operational procedures (instrumentation), 4) the approximation of our mathematical equations, and 5) the robustness of model sensitivity analysis and calibration. For example, in rainfall-runoff modeling, researchers have proposed benchmarking to examine uncertainty predictions of ML rainfall-runoff modeling (Klotz et al., 2022). For stream temperature modeling, researchers have attempted to address the role of uncertainty in deep learning model (RGCN, LSTM) prediction using the Monte Carlo Dropout (Zwart, Oliver, et al., 2023) and a unimodal mixture density network approach (Zwart, Diaz, et al., 2023).
>
> Other questions that SWT-ML studies should consider is 3) How do we make ML models generalize better, specifically with regards to ungaged basins? And 4) How can ML models be improved to predict extremes? As ML models advance to use satellite data, include more sensor networks and/or couple with climate models, there is a logical next step toward creating generalizable models that can account for extremes. In our review, only two papers by the same group (Rahmani et
al., 2020, 2023) conducted a CONUS-scale approach towards SWT-ML modeling, omitting
hydrologically important regions in the southwest (CA) and southeast (FL). Recently, a satellite
remote sensing paper used RF to model monthly stream temperature across the CONUS and tested for
temporal (walk-forward validation), unseen and 'true' ungaged regions (Philippus et al., 2024). We
have also learned that ML models such as LSTMs, generally only make predictions within the bounds
of their training data (Kratzert et al., 2019), which is a limitation for predicting extremes. Thus, we
strongly urge the community to work towards ML models that generalize better and/or are more
robust towards predictions of extremes.
Finally, 5) How can we build ML models such that they are seen as trustworthy and
interpretable by the hydrologic community? To answer this question, we must address a technical
barrier (black-box issues, data limitations, model uncertainty) and a social barrier (i.e., educated
skepticism of ML due to novelty, little understanding of computer science basics and/or coding
experience). If we are to incorporate ML into more of the decision-making process, it makes sense
that ML must be transparent and understandable to more than just computer scientists (Varadharajan
et al., 2022). For example, Topp et al. (2023) recently used explainable AI to elucidate how ML
architectures affected the SWT model's spatial and temporal dependencies, and how that in turn
affected the model's accuracy. Addressing this technical barrier can also be done by improving access
to data, which has seen remarkable progress thanks to web repositories such as NSF-funded
CUAHSI's Hydro share (CUAHSI, 2024) and GitHub (GitHub, 2024). In the United States, data
access to state and locally-based data remains limited, and should be addressed. In terms of the social
barrier, education about ML and ML-use is key. Societal interest in ML has thankfully also lead to a
plethora of educational resources and ML walk-through videos and tutorials in Tensorflow (Abadi et
al., 2015), PyTorch (Abadi et al., 2015), and Google Colab (Bison, 2019). With how fast ML-use is
evolving, short communication pieces (Lapuschkin et al., 2019) and opinion pieces (Kratzert et al.,
2024) with clear examples about an ML-issue and practical solutions could also help make ML
challenges more transparent and therefore accessible to the hydrologic community-at-large.
We have added a few lines to section 2.5 Decision Support with the provided citations:
Further focusing on the Delaware River Basin, Zwart, Oliver, et al. (2023) used data assimilation
and an LSTM to generate 1-day and 7-day forecasts of daily maximum SWT for the purpose of aiding
reservoir managers in decisions about when to release water to cool streams. Following up on this
study was Zwart, Diaz, et al. (2023), who used a LSTM and a RGCN, to generate 7-day forecasts of
daily maximum SWT for monitored and unmonitored locations in the Delaware River Basin. The
study found that the RGCN with data assimilation performed best for ungaged locations and for
higher SWT, which can serve as valuable information for reservoir operators to consider while
drafting release schedules.
**5.** In section 3 (e.g., 3.1, 3.3, 3.4), I would recommend adding some discussion regarding the equivalence
or lack of between lower-case r and r-squared, upper-case R-squared, and NSE. I am very comfortable
stating that for the purpose of this continuously valued model evaluation, upper case R-squared and NSE
are equivalent, but I am less comfortable making the assertation that lower case r and r-squared are (in all
the papers reporting this value). This is likely further complicated by the reviewed literature using the
lower-case r-squared and R-squared interchangeably, but given the 0-1 range, the high value skew, and
the special case/conditional equivalences, I believe these values should all be reported together to
characterize goodness of fit – especially that upper case R-squared and NSE should not be separated.
**AUTHOR RESPONSE**: We agree. We propose the following to address the referee's comments:
-   Revise section 3.1 text to clearly distinguish between lower-case $r$, r-squared $r^2$, and upper-case $R^2$ :
The square of r is denoted as $r^2$, or known as the square of the correlation coefficient, with values of $r^2$ ranging from 0 to 1. The $r^2$ metric is commonly used in simple linear regression to assess the
goodness of fit by measuring the fraction of the variance in one variable (i.e., observations) that can be
explained by the other variable (i.e., predictors). The metric $r^2$ tends to be confused with $R^2$, the latter
which is a statistical measure that represents the proportion of variance explained by the independent
variable(s) in a multiple linear regression model (Helsel and Hirsch, 2002). Part of the confusion may
be related to the fact that $R^2$ shares the same range of 0 to 1, with $R^2 = 1$ indicating that the model can
explain all the variance, and vice versa. We note here that while both $r^2$ and $R^2$ share similarities in that
they measure the proportion of variance, $R^2$ is more commonly used for multiple linear regression
context, while $r^2$ is best suited for simple linear regressions. To prevent confusion, we strongly suggest
that $r$, $r^2$ and $R^2$ always be reported together (even if as a supplement to a manuscript) to characterize
goodness-of-fit. The $r$ and $R^2$ metrics are typically used for normally distributed data that follows a
bivariate normal distribution (Helsel and Hirsch, 2002).

- Add text stating that upper $R^2$ and NSE should always be provided together in section 3.4:

1ˢᵗ paragraph, added after 1ˢᵗ sentence:
Having reviewed the literature and in agreement with previous published recommendations
(Moriasi et al., 2007), we recommend that a combination of standard regression (i.e., $r$, $r^2$, $R^2$),
dimensionless (i.e., NSE), and error index statistics (i.e., RMSE, MAE, PBIAS) be used for model
evaluation and reported together in future publications.

3ʳᵈ paragraph, added last sentence:
Overall, these complimentary metrics should always be reported together as they provide a
broader evaluation of model performance, i.e., NSE measures a model's predictive skill and error
variance, while $R^2$ assesses how well the model explains the variability of the data.

- In section 3.4, remove all $r^2$ values from Figure 1, only $R^2$ citations (17) remain. The median $R^2$ for
training stayed the same (0.93), while the testing $R^2$ went from 0.95 to 0.94, and the validation $R^2$ went
from 0.92 to 0.93. Overall, changes were insignificant. Below is a screenshot of the "Old (left)" and
"Revised (right)" Figure 1 for reference.

[Figure]

[Figure]

Old                                              Revised

**6.** In line 761, it feels controversial and a step too far to say ML models should be held to a higher
standard. It feels less problematic to apply these higher, seemingly attainable standards to all SWT
models. For example, a physics-based model is not "very good" by virtue of being a physics-based model,
instead it is the same "satisfactory" label because its physics are not sufficient or accurate enough to do
what the ML models can.

**AUTHOR RESPONSE**: We appreciate the referee's point of view and are open to discussion. Perhaps
instead of saying "higher standard", we can say "additional standards", but we think that additional standards are warranted nonetheless, not only in terms of performance metrics but also to improve model
transparency, eradicate black-box confusion and encourage user confidence. We disagree that a physics-
based model should be in the same "satisfactory" performance metric category because the intention of
performance metrics is to identify what fits the data best (which data-driven ML excel at), whereas the
general intention of physics-based models is to adhere to whatever governing equations have been
employed. This review shows that we have been blinded by the excellence of ML performance metrics
relative to physics-based and statistically-based models, and we need to be aware of this short sight
moving forward.
7. If possible, in addition to considering spatial extents and temporal resolution of the papers, it would be
interesting to know the aggregation level of data - if that is reported and what all the possibilities are. For
example, individual gages with input data collected at the same gage location in situ, remotely sensed data
subset to the drainage area for the reach that a gage is on. Are any works modeling dense transects along a
river or modeling raster grid cells up and across a river (i.e., the 2D surface area), etc.

**AUTHOR RESPONSE**: Thank you for the opportunity to clarify. We provided supplementary table S1
to summarize study information regarding time period, temporal resolution, spatial resolution and
hydrometeorological parameters considered by the cited studies. Responding to your comment, in our
review, we saw that the aggregation level of data is more often than not, left unreported and unclear by
studies (and reporting is not mandatory as a lot of data is pre-processed before utilization in modeling,
adding to transparency questions). We do think discerning all the possibilities of data aggregation could
make for an interesting follow-up study for the larger hydrologic community, which could focus solely on
data manipulation, processing and augmentation for ML.
**Additional literature to consider. Not necessary**

8. The paragraph at line 385 related to process guidance prompted me to recommend
https://doi.org/10.1029/2023WR035327 as very relevant. The reference is concerned with comparing
different hybrid ML methods for SWT modeling to represent groundwater processes which aren't as
represented here (e.g., relative to reservoir influence/reservoir adjacent modeling).

**AUTHOR RESPONSE:** Thank you for the suggestion, we agree that the challenge of including
groundwater influence in SWT modeling warrants more research. We want to clarify that we did not
include this reference as it appears to be a conference paper and not subjected to journal standards of peer
review. That being said, the authors of the suggested manuscript went on to publish similar work in Water
Resources Research, which we cite in this review (Topp et al., 2023).

9. In section 4.2, https://doi.org/10.1029/2020WR028091 may be a very relevant addition in-line with the
author's narrative.

**AUTHOR RESPONSE:** Thank you for the suggestion, we enjoyed reading it and think it insightful. We
added it to a proposed new 'Discussion' subsection, titled 'Future Directions of SWT Modeling', in the
first sentence (please see our response to ref #1, comment #4 for the full text):
"The utility of ML in hydrologic modeling has come a long way, with interest seemingly growing
exponentially (Nearing et al., 2021)."

**Minor writing comments:**

1.The sentence beginning on line 51 perhaps uses too bold language when stating "AI … create reasonable choices". Many users of AI and scientists have concerns regarding the reasonableness of AI. Maybe it would be more accurate to further connect with the latter part of that sentence and say that "AI … learn optimal patterns to meet stated objectives" (which may or may not be broadly reasonable)

**AUTHOR RESPONSE:** That is a good point. Reasonableness is fluid. We agree with the referee and have updated the sentence as follows:

> "Artificial intelligence (AI) describes technologies that can incorporate and assess inputs from an environment,  learn optimal patterns and implement actions to meet stated objectives or performance metrics (Xu & Liang, 2021; Varadharajan et al., 2022)."

2. Starting at line 131, "We define newer ML as those introduced in hydrologic modeling in the few years," perhaps this should say "in recent years"?

**AUTHOR RESPONSE:** We agree, thank you for the suggestion, we have updated the text to say, "in recent years".

3. At line 380, although it can be inferred, "WNN" is never explicitly defined.

**AUTHOR RESPONSE:** Thank you for catching that, we have defined the acronym.

4. At line 541, "all journals examined used least one", perhaps this should say, "at least one"

**AUTHOR RESPONSE:** Thank you! We have added the word "at".

5. By typo/mistake, it appears that two subsections in section 3 are titled "Model Performance Metrics: Error Indices"

**AUTHOR RESPONSE:** Yes, thank you for catching that mistake. Subsection 3.3 should have said "Model Performance Metrics: Dimensionless" because the subsection summarizes NSE, KGE, etc. We have updated the subsection header accordingly.

6. At line 610, there is a typo claiming an upper bound of -1

**AUTHOR RESPONSE:** Yes, that was a typo. Thank you for catching that, we have updated the text to just say "0 to 1".

7. I have the benefit of reviewing 3rd, so I read the other reviewer's comments after making my own. I agree that a characterization of the validation and test sets used would be very beneficial (e.g., spatial, temporal, spatiotemporal exclusion, etc.), but I believe the concerns of overfitting are potentially overstated by the other reviewers given that this manuscript reports train, validation, and test set metrics (and the very strong agreement between the three).

**AUTHOR RESPONSE:** Thank you for your time and energy in reviewing this manuscript. With regards to the concerns of overfitting, we include below our response to referee #1, comment #1A. We think that the referee comment with regard to "characterization of the validation and test sets" is related to referee comment #1B, which we also include below:

Section 2.4.X Overfitting and Underfitting:

When a model is too complex, i.e., has too many features or too many parameters relative to the number of observations, or is forced to overexpend its capabilities, i.e., make predictions with insufficient training data, the model runs the risk of overfitting (Srivastava et al., 2014). An overfitting model fits the training data "too well", capturing noise and details that provide high accuracy on a training dataset, only to perform poorly once the model encounters "unseen" data in testing/validation (Xu and Liang, 2021). Scenarios where overfitting may be temporarily acceptable are those where: 1) model development is at its preliminary stages, where the interest is in a "proof of life" concept, 2) when the objective is to identify heavily-relied on features by the model, i.e., feature importance, or 3) in highly-controlled modeling environments where the expected data will be consistently similar to the training dataset. The latter is more likely in certain industrial applications and unlikely in the changing nature of hydrology.

In contrast, underfitting occurs when a model is too simple to capture any patterns in the data, which can also lead to terrible performance in training, testing and validation. Underfitting can occur with inadequate model features, poor model complexity or when regularization techniques, (e.g., L1 or L2 regularization), are over-used, making the model too rigid and unable to respond to changes in the data. Given the propensity of machine learning models to effectively learn the training data, underfitting is less of an issue in ML whereas overfitting can be widespread. In the following diagram, we present an example workflow to transition away from overfitting and towards generalizability. We further encourage modelers to actively transition towards making more generalizable models, which are in theory, more capable of performing well across diverse scenarios and datasets, which will become increasingly important with the persistence of climate extremes.

**Response to ref #1, comment #1B:** We have added a few sentences (blue is new) to the Discussion subsection titled "ML as Knowledge Discovery" where we urge for TUURTs (Temporal, Unseen, Ungaged Region Tests)':

Our review finds that ML studies examining SWT have been conducted from a computational perspective, one with a focus on comparing techniques and performance metrics as opposed to explaining the nature of SWT dynamics or influencing processes. While it is understandable that not every ML-SWT paper aims to explain physical processes, we think the SWT community should come together and agree on a baseline of tests that all ML-SWT models should undergo for model robustness and transferability. Along these lines, we urge consideration of TUURTs (temporal, unseen, ungaged region tests) for future ML-SWT models as a helpful step towards not only better modeling practices but also increased model transparency and robustness. For this, we clarify that testing for "unseen" cases means testing only within the developmental dataset, whereas testing for "ungaged" cases means testing for new sites that have not been previously seen by the model at all. Recent ML-SWT studies have only applied one or two of the tests, but not all three (Topp et al., 2023; Hani et al., 2023, Souassi et al., 2023). Siegel et al. (2023), a non-ML SWT paper, tested for ungaged and unseen data but did not perform a temporal test. A relatively new study, Philippus et al. (2024), appears to be the only published SWT-ML study that purposefully applied TUURTs with some success.

*Disclaimer: I propose some additional literature (n = 4-5), and I am a coauthor on 1 of them. I do not view including that literature as mandatory, and only proposed additional sources based on their relevance to the content of this manuscript. I selected "No" to anonymity to avoid any appearance of subversive influence.*

---

## Author Response (AR1)

| 1
        | Hydrology and Earth System Science
Manuscript #HESS-2024-256
January 25 th , 2025                                                                   | 25,                                                                                                                                                                                                                     |
|-----------------------------------|------------------------------------------------------------------------------------------------------------------------------------------------------------------------------------|-------------------------------------------------------------------------------------------------------------------------------------------------------------------------------------------------------------------------|
| 5
| Subject: Response to Comments on Rev
Temperature Modeling: a review and                                                                                                         | view Paper " Machine Learning in River/Stream Water
I metrics for evaluation"                                                                                                                                 |
| 12
          | Dear Dr. Christa Kelleher, Mr. Jeremy                                                                                                                                              | Diaz and referees #1 and #2,                                                                                                                                                                                            |
| 15
| We thank you for your time and patience
line with referee feedback and think the
as our written author response to referee                                                   | ce in reviewing our manuscript. We have revised the manuscript in
e manuscript is much improved as a result. This document serves
e comments.                                                                     |
| 20
    | This document separates responses by r $21 - 30$ ). Where referee comments had both indicate that the part belonged to c all comments were kept in their original                  | referee: referee #1 (pages 2 - 14), #2 (pages 15 - 20), and #3 (pages several parts, we separated the comments into "1a, 1b, etc.," to one comment, and allow for a more organized response. Otherwise, 1 format.       |
| 24
| For revisions, new/edited text is in BLU
statement " revised lines XXX-XXX " ind
Track-Changes-Manuscript-HESS-2
manuscript " 3-Clean-Manuscript-HES | JE, while removed text is shown as being crossed out. The icates the in-line placement in the track-changes manuscript "2-024-256", where the described changes will be found. The SS-2024-256" is the "final" version. |
| 29
| Once again, thank you kindly for your t                                                                                                                                            | ime and consideration.                                                                                                                                                                                                  |
| 34
          | Sincerely,                                                                                                                                                                         |                                                                                                                                                                                                                         |
| 37
    | Cor Cor                                                                                                                                                                            | Torrie Schogue                                                                                                                                                                                                          |
| 40
| Claudia R. Corona
Postdoctoral Fellow
Colorado School of Mines                                                                                                               | Terri S. Hogue
Dean, Earth and Society Programs
Colorado School of Mines                                                                                                                                          |
| 46
          |                                                                                                                                                                                    |                                                                                                                                                                                                                         |
| 49
                |                                                                                                                                                                                    |                                                                                                                                                                                                                         |

**1 Referee #1 Comments**

The manuscript on "ML in Stream/River Water Temperature Modeling, a review and metrics for

*evaluation*" focuses on providing a comprehensive review of ML studies, including traditional and recent
methods in ML and AI, on stream temperature modeling and prediction. Overall, the manuscript is wellwritten and covers most of the relevant papers, but there are a few strategic points I would like to share

- 6 with the authors:
- 7

AUTHOR RESPONSE: We appreciate the referee's feedback and think the manuscript is much improved as a result. For reference, we separated some referee comments into a, b, etc., to provide a more
organized response. Thank you for your time and insight. Proposed new/edited text is in BLUE. Revised lines in the track-changes manuscript are indicated by the statement: (*revised lines XXX-XXX*).

**1a.** Figures 1 & 2 & 3 & table 2: The manuscript provides a table for multiple metrics such as R2, NSE, 15 RMSE, and MAE, and suggested a rate of numbers to rate the ML methods' performances. This table is 16 based on the metrics that have been achieved by the studies in the previous years which are reflected in 17 figures 1 & 2 & 3. However, those studies vary in terms of case studies, number of basins included in the study, running regional or local models. We know that ML models are prone to overfitting, especially for 18 19 stream temperature that follows a relatively sinusoidal curve through a year, which means it is more 20 predictable for complex models such as LSTM. However, it means the models are prone to easily overfit. 21 Therefore, I suggest the authors encourage the stream temperature researchers to go towards making more 22 generalizable models and less overfitted. For example, instead of suggesting performance metrics, the 23 authors can provide a few steps to make sure the models are not overfitted or underfitted. For instance, 24 always considering a spatial test on ungauged sites (basins). We know that spatial tests are more difficult 25 tasks rather than temporal tests.

26

AUTHOR RESPONSE: We agree that the SWT studies vary spatially/temporally and that ML models
risk overfitting. We appreciate the referee's comments in pointing out areas of improvement and we
suggest adding the following: 1) a new subsection 2.4.1 "SWT Predictions using ML" on
overfitting/underfitting that suggests the need for temporal- and spatially-focused testing as suggested by
the referee, and 2) a diagram showing initial steps to mitigate overfitting. The new text is below:

33

\*new Section 2.4.1, Identifying Model Complexity (revised lines 464-483)

The strong success of ML-use in SWT modeling warrants a brief and broad overview on identifying 36 model complexity to minimize overfitting and underfitting" of models. When a model is too complex, 37 i.e., has too many features or parameters relative to the number of observations, or is forced to 38 overextend its capabilities, i.e., make predictions with insufficient training data, the model runs the 39 risk of overfitting (Srivastava et al., 2014). An overfitted model fits the training data "too well", capturing noise and details that provide high accuracy on a training dataset, only to perform poorly 40 once the model encounters "unseen" data in testing/validation (Xu and Liang, 2021). Scenarios where 41 42 overfitting may be temporarily acceptable are: 1) model development is at preliminary stages, the 43 interest is in a "proof of life" concept, 2) when the objective is to identify heavily-relied on features 44 by the model, i.e., feature importance, or 3) in highly-controlled modeling environments where the 45 expected data will be consistently similar to the training dataset. The latter is more likely in industrial 46 applications and unlikely in the changing nature of hydrology.

In contrast, underfitting occurs when a model is too simple to capture any patterns in the data, which
can also lead to unsatisfactory performance in training, testing and validation. Underfitting can occur
with inadequate model features, poor model complexity or when regularization techniques, (e.g., L1
or L2 regularization), are over-used, making the model too rigid and unable to respond to changes in
the data. Given the propensity of ML models to effectively learn the training data, underfitting is less
of an issue in ML whereas overfitting can be widespread. In Figure 1, we present an example

- 1 workflow that researchers can use to transition away from overfitting and towards generalizability. In 2 the five-step outline (Fig. 1), we suggest the need for "Temporal, Unseen, Ungaged Region Tests" 3 (TUURTs), which is a call for temporal and spatially-focused testing that can be used to strengthen 4 model robustness.
  - Revised lines 484-486:

6

---

## Author Response (AR2)

Hydrology and Earth System Sciences, Manuscript #HESS-2024-256 March 23rd, 2025

Subject: Follow-up Response to Comments on Review Paper "Machine Learning in River/Stream Water Temperature Modeling: a review and metrics for evaluation"

**Dear Dr. Christa Kelleher,**

We thank you for your time and patience in handling the review of our manuscript. We also appreciate the referee's feedback and provide our text regarding the "prediction in ungaged basins" below. For revisions, new/edited text is in BLUE, removed text is <del>crossed out,</del> and original text is left in black. The statement "*revised lines XXX-XXX*" indicates the in-line placement of the described changes in "**2-Track-Changes-Manuscript-HESS-2024-256\_v3**". The document "**3-Clean-Manuscript-HESS-2024-256\_v3**" is the "final" version.

**Referee Comments from Report #1**

I would like to thank the authors for addressing the reviewers' comments. I believe they have significantly improved the manuscript. I would like to bring to the authors' attention a minor issue: the topic of 'prediction in ungaged basins' has been explored in SWT modeling for at least a decade. I encourage the authors to review the following two papers, as well as the references cited within them, which may help provide additional context and background: <a href="https://doi.org/10.5194/hess-19-3727-2015">https://doi.org/10.5194/hess-19-3727-2015</a> and <a href="https://doi.org/10.1002/hyp.14400">https://doi.org/10.1002/hyp.14400</a>

**AUTHOR RESPONSE:** We agree that the topic of 'prediction in ungaged basins' has been previously explored and we appreciate the opportunity to clarify. The added text is below (*revised lines 894-914*) and includes the references suggested by the reviewer (in **bold**):

The challenge of prediction in ungaged basins in SWT modeling has been explored for at least a decade by processbased (Dugdale et al., 2017) and statistically based (Gallice et al., 2015, Isaak et al., 2017; Wanders et al., 2019; Siegel et al., 2023) models. Unfortunately, process-based models continue to be limited by data requirements and memory or processing/programming impediments (Dugdale et al., 2017; Ouellet et al., 2020), while statistically based models struggle to account for changing physical conditions (Benyahya et al., 2007; Arismendi et al., 2014; Lee et al., 2020). Physics-derived statistically based models have been applied in ungaged regions (Gallice et al., 2015) but models tend to be region-specific and not generalizable. We posit that a future direction of ML models is to expand on their ability to learn, identify and mimic the complexity needed to improve SWT predictions for ungaged basins. To date, researchers have used ML to model SWT for partially ungaged (i.e., discharge used as input) regions across the CONUS (Rahmani et al., 2020, 2021), though limitations persist in In our review, only two papers by the same group (Rahmani et al., 2020, 2023) conducted a CONUSscale approach towards SWT-ML modeling, omitting hydrologically important complex and critical regions in the southwest (CA) and southeast (FL). Recently, a satellite remote sensing paper used RF to model monthly stream temperature across the CONUS and tested for temporal (walk-forward validation), unseen and 'true' ungaged regions (Philippus et al., 2024). Given community-wide modeling interest expanding from SWT prediction to forecasting (Zhu and Piotrowski, 2020; Jiang et al., 2022; Zwart, Diaz, et al., 2023), ML-use could prove essential in capturing unknown, complex SWT patterns in space and time (Philippus, Corona, et al., 2024) and with shifting baselines. We have also learned that With regards to ML models such as LSTMs predicting extremes, a limitation that must be addressed with ML models such as LSTMs, is that they generally only make predictions within the bounds of their training data (Kratzert et al., 2019) though researchers are looking to improve on this by using ML-hybridizations (Rozos et al., 2023). , which is a limitation for predicting extremes. Thus, we strongly urge Overall, there is promising work in the community towards creating ML models for SWT that generalize better and/or are more robust towards for predictions of extremes.

**Additionally, we describe the Rahmani et al. (2021) study (revised lines 539-542):**

A follow-up study by **Rahmani et al. (2021)** used six years of SWT data and relevant meteorological parameters for 455 sites across the CONUS (minus California and Florida) to test LSTM models for data-scarce, dammed, and semi-ungaged basins (discharge used as input). The follow-up study showed improved performance, but the LSTM models remained limited in capturing the influence of latent contributions such as base-flow and subsurface storage.

We updated our calculations of performance metrics (screenshots below) to include the suggested Rahmani et al. (2021), for NSE (top, fig.3) and RMSE (bottom, fig.4). We note no significant changes (*revised lines 730-735*):